# Transcription factor Emx2 controls stereociliary bundle orientation of sensory hair cells

Tao Jiang[1,2], Katie Kindt[3]*, Doris K Wu[2]*

[1]Program in Neuroscience and Cognitive Science, University of Maryland, College Park, United States; [2]Laboratory of Molecular Biology, National Institute on Deafness and Other Communication Disorders, National Institutes of Health, Bethesda, United States; [3]Section on Sensory Cell Development, National Institute on Deafness and Other Communication Disorders, National Institutes of Health, Bethesda, United States

**Abstract** The asymmetric location of stereociliary bundle (hair bundle) on the apical surface of mechanosensory hair cells (HCs) dictates the direction in which a given HC can respond to cues such as sound, head movements, and water pressure. Notably, vestibular sensory organs of the inner ear, the maculae, exhibit a line of polarity reversal (LPR) across which, hair bundles are polarized in a mirror-image pattern. Similarly, HCs in neuromasts of the zebrafish lateral line system are generated as pairs, and two sibling HCs develop opposite hair bundle orientations. Within these sensory organs, expression of the transcription factor Emx2 is restricted to only one side of the LPR in the maculae or one of the two sibling HCs in neuromasts. Emx2 mediates hair bundle polarity reversal in these restricted subsets of HCs and generates the mirror-image pattern of the sensory organs. Downstream effectors of Emx2 control bundle polarity cell-autonomously via heterotrimeric G proteins.

*For correspondence: katie. kindt@nih.gov (KK); wud@nidcd. nih.gov (DKW)

**Competing interests:** The authors declare that no competing interests exist.

## Introduction

The asymmetric location of cilia on the apical surface of epithelial cells is critical for many functions such as left-right asymmetry and normal flow of the cerebral spinal fluid. Abnormal positioning of the monocilium in the node and multicilia of ependymal cells lining the brain ventricles can lead to left-right asymmetry defects and hydrocephalus, respectively (*Tissir et al., 2010*; *Song et al., 2010*). Similarly, the asymmetrical orientation or polarity of hair bundle on top of sensory HCs provides the directional sensitivity for detecting sensory inputs in the form of vibrations (*Shotwell et al., 1981*). The mechanisms underlying the precise positioning of hair bundle are unclear.

Each hair bundle is comprised of specialized microvilli (also called stereocilia) arranged in a staircase pattern that are tethered to the kinocilium, a true cilium. Deflection of the hair bundle towards its kinocilium leads to the opening of the mechanotransduction channels located on the tips of stereocilia and depolarizes the HC. Hair bundle displacement in the opposite direction results in hyperpolarization (*Figure 1A*; *Shotwell et al., 1981*). Therefore, the orientation and positioning of the hair bundle determine the directional sensitivity of the HC.

Within each sensory organ of the inner ear, HCs display a well-defined pattern of hair bundle polarity tailored to its function. The organ of Corti (sensory organ for sound detection) and the three cristae (vestibular organs for detecting angular acceleration) exhibit a polarity pattern where all the hair bundles are polarized in the same direction (unidirectional). In contrast, maculae of the utricle and saccule, vestibular organs for detecting linear acceleration, are divided by the LPR into two

regions with opposing hair bundle polarities. Across the LPR, hair bundles are oriented toward each other in the utricle and away from each other in the saccule (*Figure 1A*). Similar to the hair bundle polarity pattern in the utricle, HCs in neuromast organs of the zebrafish lateral line are also oriented toward each other in a 1:1 ratio (*López-Schier et al., 2004*).

Asymmetric localization of the hair bundle begins with the docking of the basal body to the center of the apical surface to form the kinocilium. The nascent kinocilium then moves from the center to the periphery where adjacent specialized-microvilli gradually differentiate into a staircase architecture and establishes the intrinsic polarity of the hair bundle (*Lu and Sipe, 2016*; *Tilney et al., 1992*). The directed movement of the kinocilium and subsequent staircase formation of the stereocilia in each HC require an intracellular complex, Insc/LGN/Gαi, which has been determined to mediate spindle orientation during mitosis in other cell types (*Morin and Bellaïche, 2011*; *Lu and Sipe, 2016*; *Ezan et al., 2013*; *Tarchini et al., 2013*, *2016*). This complex, tethered to the cell cortex of sensory HCs, presumably relocates the basal body/nascent kinocilium to the cell periphery via the attached microtubules, in a manner similar to relocating centrioles during mitosis. In humans, mutations of LGN (also known as GPSM2) cause a syndromic hearing loss suggesting that the directed kinocilium migration and bundle assembly are critical for hearing (*Doherty et al., 2012*).

In addition to establishing the intrinsic polarity of individual HCs, precise arrangement of hair bundle patterns in each inner ear organ is regulated by global and intercellular signaling mechanisms as well. The global polarity signaling molecules Wnts are required for establishing directional polarity in the developing mouse limb bud and *Drosophila* wing (*Wu et al., 2013*; *Gao et al., 2011*), as well as hair bundle alignment in the organ of Corti (*Qian et al., 2007*; *Dabdoub and Kelley, 2005*). A highly conserved intercellular signaling pathway, the core planar cell polarity (cPCP) pathway, coordinates polarity between adjacent cells. This cPCP complex is comprised of transmembrane proteins Van Gogh (Vang), Frizzled (Fz), and Celsr as well as cytoplasmic proteins such as Dishevelled (Dvl) and Prickle (Pk) (*Goodrich and Strutt, 2011*; *Wallingford, 2012*). Mutations of cPCP components such as Van Gogh-like 2 (Vangl2), Fz, and Dvl, affect the coordinated alignment of hair bundles among HCs but not the peripheral positioning of the kinocilium or the intrinsic bundle polarity of individual HCs (*Montcouquiol et al., 2003*; *Wang et al., 2005*, *2006*).

Hair bundles are misaligned in the maculae of *Vangl2* mutants and the LPR is disrupted (*Montcouquiol et al., 2006*; *Yin et al., 2012*). However, it is debatable whether this complex is directly involved in the LPR establishment because the distribution of cPCP components such as Prickle-like 2 (Pk2) and Vangl2 are not reversed across the LPR in normal maculae as predicted by the polarity of the hair bundle (*Deans et al., 2007*; *Jones et al., 2014*). By contrast, all hair bundles were orientated in the same direction and the LPR was reportedly absent in knockout maculae of *Emx2,* which encodes a homeobox-containing transcription factor (*Holley et al., 2010*). This defect was attributed to an imbalance between proliferation and differentiation of sensory precursors rather than a specific role for Emx2 in directing hair bundle polarity.

In our study, we show that *Emx2* expression is required to reverse hair bundles from their 'default' polarity in the macula and neuromast. Ectopic expression of *Emx2* in HCs is sufficient to reverse the 'default' bundle polarity. Our loss- and gain-of-function experiments in both inner ear and neuromast indicate that Emx2 is necessary and sufficient to mediate hair bundle orientation in vertebrate HCs.

## Results

### The border of *Emx2* expression domain coincides with the LPR in maculae

We investigated the reported hair bundle polarity phenotype in *Emx2* knockout maculae (*Holley et al., 2010*) by examining the normal expression pattern of Emx2. Using both in situ hybridization and immunostaining, we found that Emx2 expression is restricted to the lateral extrastriolar region (LES) of the utricle and inner region (IR) of the saccule (*Figure 1B,C,C',F,F'*, *Figure 1—figure supplement 1A–F*). The border of the Emx2 expression domain (*Figure 1C',F'*, green line bordering the green region) closely follows the LPR in the two maculae, at the lateral edge of the oncomodulin-positive striola in the utricle and at the center of the saccule bisecting the striola (blue outlined; *Li et al., 2008*; *Desai et al., 2005*). The coincidence of the Emx2 border (green line) with the LPR

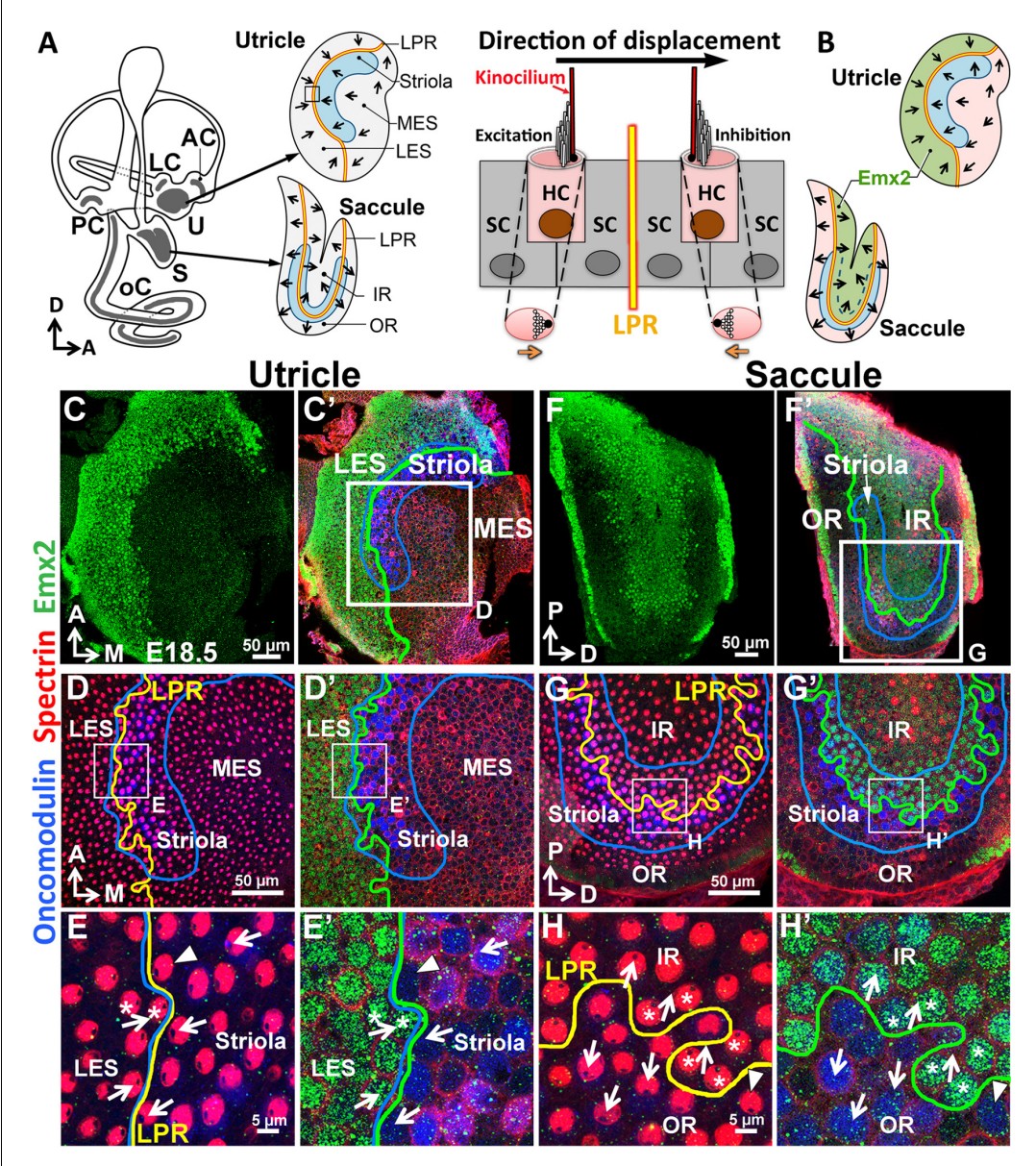

**Figure 1.** Regional expression of *Emx2* in the maculae. (**A**) Medial view of a mouse left inner ear with its six sensory organs (grey). Enlarged are the utricle and saccule showing their subdivisions, hair bundle polarity (arrows), LPR (yellow line), and striola (blue). The square denotes the sensory epithelium across the LPR on the right. Displacement of the hair bundle towards or away from the kinocilium results in depolarization and hyperpolarization of HC, respectively. (**B**) Schematic of the Emx2 expression domain (green) in the utricle and saccule. (**C–H'**) Utricle (**C–E'**) and saccule (**F–H'**) are stained for anti-Emx2 (green), anti-spectrin (red) and anti-oncomodulin (blue) antibodies (n = 6). Hair bundle polarity is determined by the location of the kinocilium, which is devoid of anti-spectrin staining (red). (**D,E,G,H**) and (**D',E',G',H'**) are confocal images of the same HCs taken at the apical surface and nuclei level, respectively. (**C,C',D',E'**) Anti-Emx2 (green) staining is located in the lateral extrastriola (LES), which is lateral to the oncomodulin-positive striola (blue) of the utricle. Hair bundles in LES point toward those in the striola and medial extrastriola (MES) of utricle (**E**). (**F,F'**, **G',H'**) Anti-Emx2 staining is restricted to the inner region (IR) of saccule and the LPR bisects the striola. Hair bundles in the IR point away from those in the outer region (OR) of saccule (**H**). The border of the Emx2-positive domain (green line, **D',E',G',H'**) coincides with LPR (yellow line, **D,E,G,H**) in both maculae. Refer to *Figure 2* for a description of asterisks and arrowheads. A, anterior; AC, LC, and PC, anterior, lateral and posterior crista; D, dorsal; M, medial; oC, organ of Corti; P, posterior; S, saccule; U, utricle.

The following figure supplement is available for figure 1:

**Figure supplement 1.** Expression of Emx2 in sensory organs of the mouse inner ear.

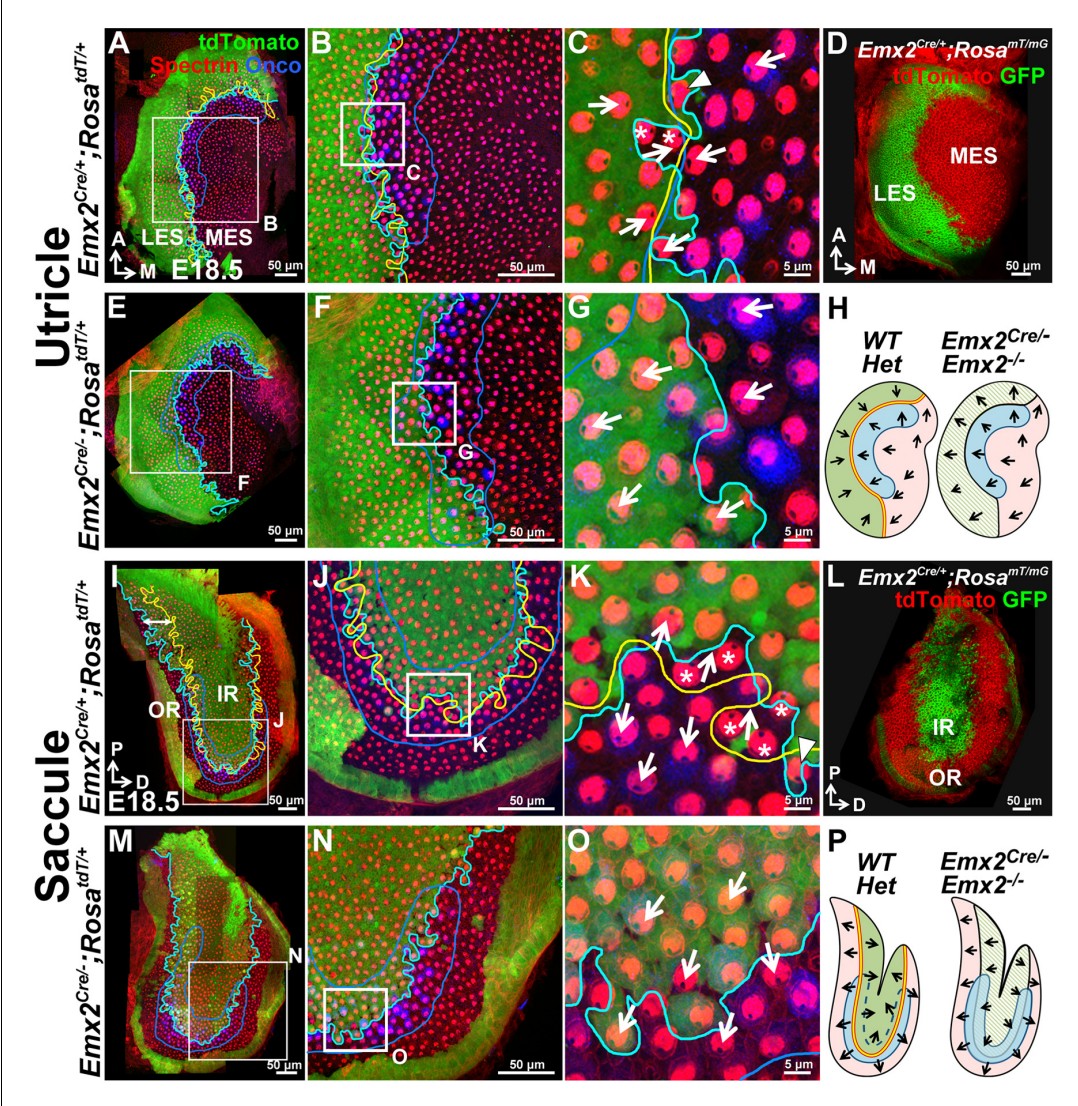

**Figure 2.** The Emx2 lineage domain is present in *Emx2* functional null maculae. (A–C) and (I–K) are the same specimens as (C–E′) and (F–H′) in *Figure 1*. (A–C) In *Emx2*<sup>cre/+</sup>;*Rosa*<sup>tdT/+</sup> utricles, the border of the *Emx2* lineage domain (cyan line) coincides largely with LPR (yellow line), located at the lateral edge of the oncomodulin-positive striola (blue outlined; n=5). (C) HCs point toward each other (arrows) across the LPR. Asterisks label HCs that are negative for cre reporter signals but positive for Emx2 immunostaining (*Figure 1E′*; n=18), whereas arrowhead labels a HC that is cre-reporter positive but negative for Emx2 immunostaining (*Figure 1E′*; n=21). (D) A similar *Emx2* lineage domain (GFP) is observed using a different Cre reporter, *Rosa*<sup>mT/mG</sup>. (E–G) The *Emx2* lineage domain (green) remained in *Emx2*<sup>Cre/-</sup>;*Rosa*<sup>tdT/+</sup> utricles, but hair bundle polarity in this region is reversed (G; n=5) compared to controls (C; n=5). (I–K) In *Emx2*<sup>Cre/+</sup>;*Rosa*<sup>tdT/+</sup> control saccules, the border of the *Emx2* lineage domain (cyan line) mostly coincides with the LPR except in the ventral-posterior region (I, double-headed arrow). (K) Across the LPR, HCs are pointing away from each other (arrows). Asterisks label HCs that are negative for cre reporter signals but positive for Emx2 immunostaining (*Figure 1H′*; n=60), whereas arrowhead labels a HC that is cre-reporter positive but negative for Emx2 immunostaining (*Figure 1H′*; n=105). (L) A similar lineage domain (GFP) is observed using the Cre reporter *Rosa*<sup>mT/mG</sup>. (M–O) In *Emx2*<sup>Cre/-</sup>;*Rosa*<sup>tdT/+</sup> mutant saccules, the border of the *Emx2* lineage domain (cyan line) remained located in the middle of the striolar region (blue outline), but the hair bundles within the lineage domain are reversed (O; n=5), relative to controls (green region in K). (H) and (P) Schematics of respective utricles and saccules showing relationships among the *Emx2* expression domain (green), hair bundle polarity pattern, LPR (yellow line) and striola (blue outlined) in controls and *Emx2* mutants.

The following source data and figure supplements are available for figure 2:

**Source data 1.** Sensory area quantifications for *Emx2* mutant maculae.
**Source data 2.** Total hair cell number in utricles of loss- and gain-of-function *Emx2* mutants.

*Figure 2 continued on next page*

*Figure 2 continued*

**Figure supplement 1.** Emx2 immunostaining in the *Emx2^Cre/+* maculae correlates with the reversed hair bundle polarity.

**Figure supplement 2.** Mutant phenotypes of *Emx2^Cre/Cre* and *Emx2^Cre/-* embryos are similar to *Emx2 ^-/-* mutants.

(yellow line) was confirmed by comparing Emx2 immunoreactivity to hair bundle polarity. We used the absence of anti-spectrin staining at the kinocilium location for determining hair bundle polarity (*Figure 1D–E',G–H'*; *Deans et al., 2007*). Using this approach, we found that only HCs lateral to the LPR in the utricle and internal to the LPR in the saccule are Emx2 positive and they show polarities different from the rest of the maculae (*Figure 1D–E',G–H'*, arrows). These results indicate that *Emx2* expression is restricted to one side of the LPR in the maculae that share the same hair bundle polarity.

## The LES and IR regions are preserved in *Emx2^-/-* maculae

Extrapolating from the normal restricted expression of *Emx2* shown above and the unidirectional polarity phenotype previously reported in *Emx2* knockouts (*Holley et al., 2010*) suggest that *Emx2* has a role in regional hair bundle polarity patterning. However, since *Emx2* encodes a transcription factor and has been implicated in regional patterning in other systems (*Pellegrini et al., 1996*; *Miyamoto et al., 1997*), the lack of LPR in *Emx2^-/-* maculae could be caused by the loss of the *Emx2* expression domain rather than altering the hair bundle polarity. Indeed, the area of the maculae was reported to be smaller in *Emx2^-/-* mutants than wildtype (*Holley et al., 2010*). To address this possibility, we took a genetic approach. First, we lineage-traced the descendants of *Emx2* expressing cells in wildtype by crossing *Emx2^Cre/+* mice (*Kimura et al., 2005*) to *Cre* reporter mice, *Rosa^tdT/+* or *Rosa^mT/mG*. The lineage-traced domain of *Emx2* (*Figure 2A–D,I–L*) corresponded to its expression pattern (*Figure 1C–H'*), regionally restricted to the LES of utricle and IR of saccule. Within the *Emx2*-lineage domain of the two maculae (*Figure 2B–C,J–K*, green), hair bundles were oriented in the opposite direction from the rest of the sensory organ. The border of the lineage domain in the maculae (*Figure 2A–C,I–K*, cyan line) largely coincided with the LPR (yellow line). Immediately lateral or inside of the LPR in the respective utricle and saccule, there were occasional cre reporter-negative HCs with reversed polarity (*Figure 2C,K*, *Figure 2—figure supplement 1*, asterisks) but these cells were invariably positive for Emx2 immunostaining (*Figure 1E',H'*, *Figure 2—figure supplement 1*, asterisks). By contrast, medial or outside of the LPR in the respective utricle and saccule, there were also cre-reporter positive HCs with default polarity (*Figure 2C,K*, *Figure 2—figure supplement 1*, arrowheads) but they were negative for Emx2 immunostaining (*Figure 1E'H'*, *Figure 2—figure supplement 1*, arrowheads). These cre-reporter positive HCs with default polarity were particularly abundant in the posterior-ventral region of the saccule where the border of *Emx2*-lineage domain and LPR diverged (*Figure 2I*, double-headed arrow). Overall, these results indicate that the border of the *Emx2*-lineage domain, though not as faithful as the immunostaining results, corresponds reasonably well with the LPR.

Encouraged by the lineage results, we investigated whether *Emx2^Cre*, in which *Cre* is inserted into exon1 of the *Emx2* locus, generates a functional null. Our results showed that *Emx2^Cre/Cre* and *Emx2^Cre/-* embryos exhibit brain, kidney and ear phenotypes similar to *Emx2^-/-* mutants indicating that *Emx2^Cre* is a null allele (*Figure 2—figure supplement 2*; *Pellegrini et al., 1996*; *Miyamoto et al., 1997*). Thus, this *Emx2^Cre* strain allowed us to investigate the *Emx2*-lineage domain in *Emx2* functional null mutants. We observed that the *Emx2*-lineage domain remained in both maculae of *Emx2^Cre/-;Rosa^tdT/+* mutant ears. The spatial relationship of *Emx2*-lineage domain with the striola was maintained but hair bundle polarity within the domain was reversed (*Figure 2E–H,M–P*, green region) compared to controls (*Figure 2C,K*, green region). We also did not observe a reduction in the size of maculae (*Figure 2—source data 1*) or total HC number in the utricle (*Figure 2—source data 2*) of our *Emx2* mutants. Taken together, our results indicate that the unidirectional hair bundle phenotype in *Emx2^-/-* maculae is not caused by the loss of a domain, rather Emx2 has a role in establishing regional bundle polarity in the maculae. Furthermore, the strong correlation between the

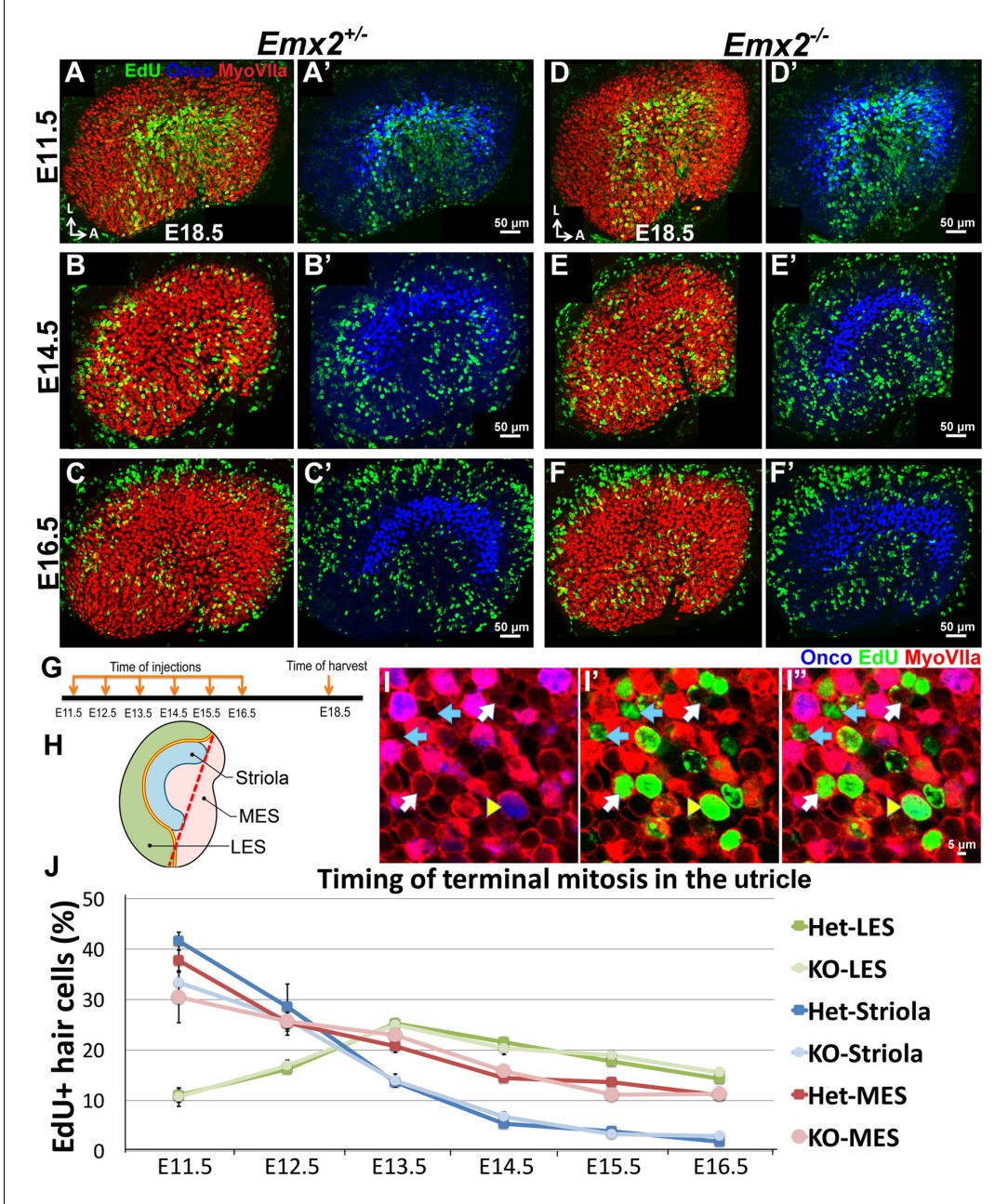

**Figure 3.** Regional differences in timing of cell cycle exit are maintained in the *Emx2⁻/⁻* utricle. (A–C') E18.5 *Emx2⁺/⁻* utricles injected with EdU at E11.5 (A-A'; n = 3), E14.5 (B-B'; n = 3), or E16.5 (C-C'; n = 3) and harvested at E18.5 (G) were processed for anti-Myosin VIIa (HC marker), anti-oncomodulin (striolar marker), and incorporated EdU. Abundant labeling is observed in the striola when the utricle is exposed to EdU at E11.5 (A) and not at E14.5 (B) and E16.5 (C), whereas labeling in the MES is observed in all three ages but weakest at E16.5 (C). In contrast, EdU labeling is not apparent in the LES at E11.5 (A) but only at E14.5 and E16.5 (B,C). (D–F') In *Emx2⁻/⁻* utricles, the regional differences of EdU labeling are similar to the controls. (H) A subdivision of the utricle into the oncomodulin-positive, striola (blue), LES (green) and MES (pink) domains. The red dotted line joining the two ends of the striola was used as an arbitrary division between the two ends of LES and MES. (I–I") Triple labeling of the utricle with anti-Myosin VIIa (red), anti-oncomodulin (blue) and EdU (green). HCs that are triple-labeled (yellow arrowhead), EdU-positive HCs (white arrow) and EdU-positive non-HCs (blue arrow) are marked. (J) No significant differences in percentages of EdU-labeled HCs are observed between various regions of the *Emx2⁺/⁻* and *Emx2⁻/⁻* utricles (*Figure 3—source data 1*). Blue, green and pink lines represent the percentages of EdU-positive HCs in the striola, LES, and MES, respectively. Both *Emx2* heterozygous and knockout utricles show the peak of EdU-positive HCs detected at E11.5 for the striola and MES, and at E13.5 for the LES.

The following source data is available for figure 3:

**Source data 1.** Timing of terminal mitosis in utricular HCs of *Emx2* mutants.

border of Emx2 immunostaining domain and the LPR suggests that Emx2 has a cell-autonomous role in reversing bundle polarity.

## Timing of terminal mitosis of HC precursors in the *Emx2*[-/-] utricle is unchanged

Since *Emx2* has been implicated in regulating cell divisions in the brain (*Galli et al., 2002*; *Heins et al., 2001*), hair bundle polarity defects in the *Emx2* mutants could be indirectly related to the timing of terminal mitosis in the maculae. For example, sensory progenitors expressing Emx2 may remain in the cell cycle longer and fail to respond to polarity signal(s) that are instructing post-mitotic HCs in non-Emx2 regions. We addressed this possibility by comparing the timing of HC terminal mitosis between *Emx2* heterozygous and knockout utricles. A thymidine analog, EdU, was injected intraperitoneally to pregnant dams between embryonic day (E) 11.5 to E16.5, and embryos were harvested at E18.5. Myosin VIIa-positive HCs with strong EdU labeling served as a proxy for the time of terminal mitosis. Although our results indicate that HC precursors in the LES (where Emx2 is normally expressed) undergo terminal mitosis later than the rest of the maculae, we did not observe any obvious difference in this timing between controls and *Emx2*[-/-] utricles (*Figure 3*). These results suggest that *Emx2* does not affect hair bundle polarity indirectly via regulating the timing of terminal mitosis of HC precursors in the LES.

## Ectopic *Emx2* reverses hair bundle polarity

If Emx2 directs hair bundle polarity, one would predict the ectopic expression of *Emx2* in naïve (non-*Emx2*) HCs should alter their polarity. Thus, we generated *Rosa*[Emx2] mice, modeled after the Cre reporter, *Rosa*[tdT/+] (Rosa-Ai14; *Madisen et al., 2010*), such that *Emx2* transcription can be activated in the presence of Cre. We first ectopically expressed *Emx2* in all sensory epithelia of the inner ear by breeding *Sox2*[CreER] (*Arnold et al., 2011*) to *Rosa*[Emx2] mice and administered tamoxifen to pregnant dams at E12.5/E13.5 (GOF-SE early). Compared to hair bundles pointing toward the LPR in control utricles (*Figure 4A–C*), the LPR was absent in GOF-SE early mutants. All hair bundles in the entire utricle were uniformly pointing toward the medial direction (*Figure 4D–F,O–Q*). This ability of ectopic *Emx2* to reverse hair bundle polarity is developmentally dependent. By delaying the tamoxifen administration to E15.5 and E16.5 (GOF-SE late), at which time many HC precursors have undergone terminal mitosis and started to differentiate (*Figure 3*), only a partial polarity phenotype was observed in the medial utricle (*Figure 4G–I,P–Q*, region M). These results suggest that once the kinocilium position is established, it no longer responds to Emx2-dependent cues. Ectopic expression of *Emx2* during early prosensory development also resulted in increased apical HC surface area and reduced HC density, which may not be related to the bundle polarity phenotype (*Figure 4D–F, M–O*). To better pinpoint the role of *Emx2* in hair bundle polarity, we overexpressed *Emx2* specifically in nascent HCs rather than all cell types in the prosensory domain using the *Gfi1*[Cre]strain (*Yang et al., 2010*). In *Gfi1*[Cre/+];*Rosa*[Emx2/+] mutant utricles (GOF-HC), all HCs in the utricle were pointing towards the medial edge (*Figure 4J–L,P–Q*) even though the total HC number remained the same (*Figure 2—source data 2*). The hair bundle reversal in the medial utricle was approximately 180° different from controls (*Figures 4P* and 8K, compare region M between control and GOF-HC utricles). Similar findings were observed in the saccule: only the outer region (OR) of the saccule where *Emx2* is not normally expressed showed a hair bundle reversal phenotype (*Figure 5C–D,O*) compared to controls (*Figure 5A–B*). Gain-of *Emx2* function in naïve HCs also affected some of their endogenous properties since only a low percentage of *Gfi1*[Cre/+];*Rosa*[Emx2/+] maculae retained their oncomodulin staining (*Figure 5C–D*, n = 1/5).

In addition to maculae, we also examined the impact of Emx2 in other sensory organs. In the vestibular system, hair bundles in cristae detect unidirectional fluid flow through semicircular canals and all three cristae exhibit a defined polarity. For example, hair bundles are all oriented toward the anterior direction in the anterior crista and medial direction in the lateral crista (*Figure 5E–G*, arrows), and Emx2 immunostaining was not detected in these HCs (*Figure 1—figure supplement 1G–H'*; *Holley et al., 2010*). In *Gfi1*[Cre/+];*Rosa*[Emx2/+] ears, hair bundle polarity in both anterior and lateral cristae was reversed (*Figure 5H–J,O*) compared to controls (*Figure 5E–G*). By contrast, in the organ of Corti, Emx2 is normally expressed in the HCs as well as the Hensen's and Claudius' cells (*Figure 1—figure supplement 1I–J'*; *Holley et al., 2010*). As a result, no hair bundle polarity

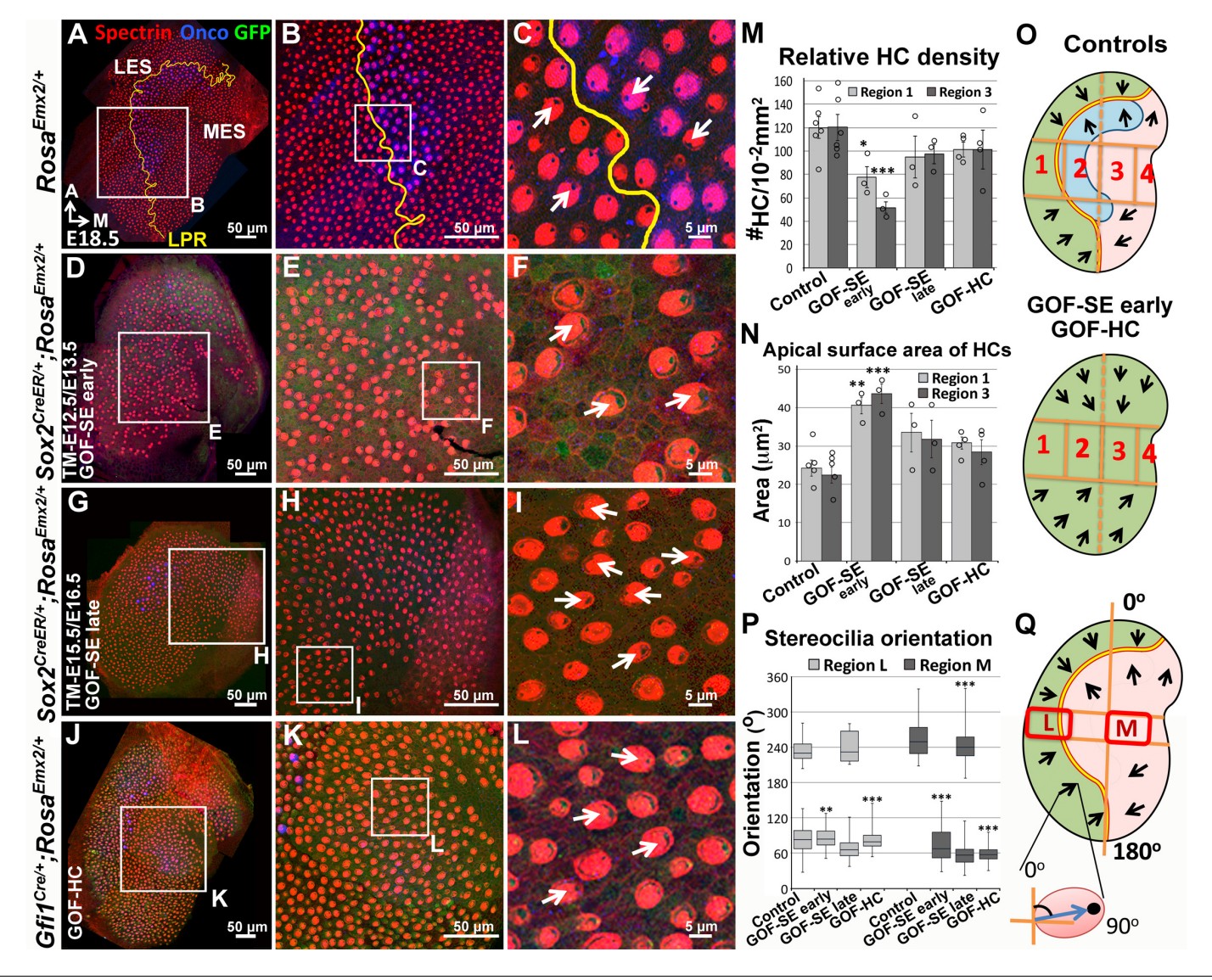

**Figure 4.** Ectopic *Emx2* reverses hair bundle polarity in the utricle. Compared to the medial side of the LPR in controls (**A–C**), hair bundle polarity (arrows) in the medial region of *Sox2*^*CreER/+*^;*Rosa*^*Emx2/+*^ (**D-F**, tamoxifen given at E12.5 and E13.5, gain-of-function (GOF) in the sensory epithelium (SE), GOF-SE early; n = 3) and *Gfi1*^*Cre/+*^;*Rosa*^*Emx2/+*^ utricles (**J-L**, GOF-HC (hair cell); n = 7) are reversed. (**G–I**) *Sox2*^*CreER/+*^;*Rosa*^*Emx2/+*^ utricles (tamoxifen give at E15.5 and E16.5, GOF-SE late) show a mixed phenotype of normal and reversed bundle polarities (white arrows, n = 3). Among the GOF specimen (**D,G,J**), anti-oncomodulin staining (blue) is only sparsely detectable in the GOF-HC utricles (**J**) compared to controls (**A**). (**M–N**) Quantification of HC density (**M**; *Figure 4—source data 1*) and surface area (**N**; *Figure 4—source data 2*) in Regions 1 and 3 of utricles from various genotypes (controls, n = 6; GOF-SE, n = 3; GOF-HC, n = 4). Only utricles of GOF-SE early show a significant decrease in HC density and an increase in the apical surface area of HCs compared to controls. Error bars represent SEM. *p<0.05; **p<0.01; ***p<0.001. (**O**) Schematic diagrams illustrating hair bundle polarity pattern and defined regions in control and *Emx2* gain-of-function utricles. (**P–Q**) Box-plots of hair bundle polarity in regions L (LES) and M (MES) of utricles (*Figure 4—source data 3*). Region L of controls includes the LPR and thus show HCs with both medial and lateral polarities but only medial-pointing hair bundles are present in all *Emx2* GOF utricles except GOF-SE late (control, n = 6; mutant, n = 3). Box represents quartiles 1 to 3. The line within the box is the median and the bar represents maximum and minimum number. **p<0.01, ***p<0.001, compared to controls.

The following source data is available for figure 4:

**Source data 1.** Quantification of HC density in *Emx2* gain-of-function mutants.
**Source data 2.** Quantification of apical surface area of HCs in *Emx2* gain-of-function mutants.
**Source data 3.** Quantification of hair bundle orientation in *Emx2* gain-of-function mutants.

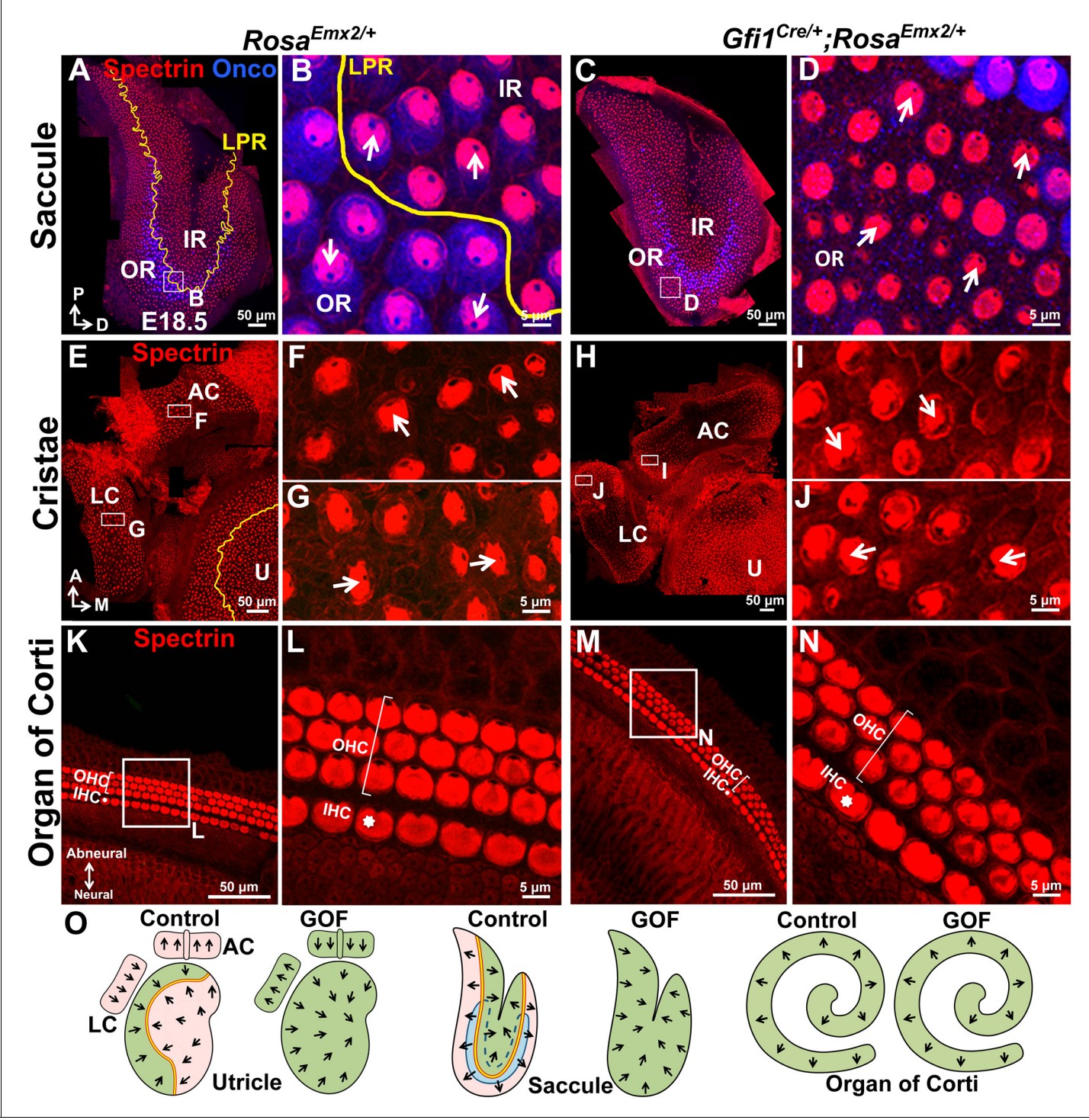

**Figure 5.** Ectopic *Emx2* reverses hair bundle polarity in the saccule and cristae but not organ of Corti. (A–D) Saccule, (E–J) anterior and lateral cristae (AC and LC), as well as (K–N) organ of Corti of $Rosa^{Emx2/+}$ (A–B, E–G, K–L) and $Gfi1^{Cre/+};Rosa^{Emx2/+}$ (C–D, H–J, M–N) inner ears at E18.5 (n = 7). In the $Gfi1^{Cre/+};Rosa^{Emx2/+}$ saccule, bundle polarity in the OR (outer region) is reversed by approximately 180° (C–D) compared to controls (A–B). (E–J) In control cristae, hair bundles point toward the anterior direction in the AC (F) and medial direction in the LC (G), whereas these polarities are reversed in AC (I) and LC (J) of $Gfi1^{Cre/+};Rosa^{Emx2/+}$ ears, pointing toward posterior and lateral directions, respectively (H–J). (K–N) No difference in hair bundle polarity is apparent between control (K–L) and $Gfi1^{Cre/+};Rosa^{Emx2/+}$ (M–N) organ of Corti. IHC, inner HCs (asterisk); OHC, outer HCs (bracket). (O) Schematic diagrams illustrating hair bundle polarity pattern in control and *Emx2* GOF sensory organs of the inner ear.

abnormality was observed in the cochlea of *Gfi1^{Cre/+};Rosa^{Emx2/+}* mutants (*Figure 5K–O*), similar to the LES and IR. Consistent with the normal expression pattern of *Emx2* and these GOF results, *Emx2^{-/-}* mutants have no apparent phenotype in the cristae but the outer HCs are absent and inner HCs are poorly organized into two rows (*Figure 2—figure supplement 2H–K*; *Holley et al., 2010*). We attributed these cochlear phenotypes to an earlier requirement of *Emx2* function in the organ of Corti prior to kinocilium positioning.

The *Gfi1^{Cre/+};Rosa^{Emx2/+}* mice survived until early postnatal ages and they exhibited balance deficits that are presumably caused by the polarity phenotype in the cristae and maculae. Taken together, these results suggest that *Emx2* has a dominant, cell-autonomous effect in dictating hair bundle polarity pattern in the inner ear which is likely to be important for normal inner ear functions.

## Distribution of intercellular and intracellular polarity components in *Emx2* mutants

To determine how downstream effectors of Emx2 mediate hair bundle orientation, we investigated whether an intercellular polarity pathway, cPCP, responsible for hair bundle alignment was altered in *Emx2* mutants. Comparing the distribution of some of the cPCP proteins such as Pk2 and Vangl2 between control and *Emx2* mutant utricles, we confirmed that Pk2 immunoreactivity in control utricles is always concentrated on the medial side of the HC-supporting cell border despite the hair bundle polarity reversal across the LPR (*Figure 6A–C*; *Deans et al., 2007*). This expression pattern of Pk2 is maintained in both loss- and gain-of *Emx2* function utricles, including the respective lateral and medial regions of mutants where the hair bundle polarity is reversed (*Figure 6D–G*, *Figure 2—figure supplement 2F–G*). Similar to Pk2, the distribution of Vangl2 in control utricles, though strongest between supporting cells, is not changed across the LPR (*Figure 6—figure supplement 1A–B*; *Jones et al., 2014*). This expression pattern of Vangl2 is also maintained in the *Emx2* mutants (*Figure 6—figure supplement 1C–F*). Together, these results indicate that the distribution of intercellular polarity proteins in the utricle is not altered by loss- or gain- of *Emx2* function and suggest that *Emx2* functions independently or downstream of the cPCP complex.

The intracellular signaling complex, Insc/LGN/Gαi, is required to guide the migration of the kinocilium to its asymmetrical apical location in HCs (*Ezan et al., 2013*; *Tarchini et al., 2013*). This complex is distributed as a crescent shape that is always associated with the kinocilium at the apical surface of HCs (*Ezan et al., 2013*; *Tarchini et al., 2013*). In other systems, the Insc/LGN/Gαi complex is often associated with one of the members of Par membrane proteins (*Di Pietro et al., 2016*). By contrast, a Par protein, Par6, is found complementary to the Insc/LGN/Gαi complex in HCs and located opposite the kinocilium (*Figure 6—figure supplement 1G–H*; *Ezan et al., 2013*). The distribution patterns of Gαi and Par6 are reversed across the LPR, in alignment with the position of the kinocilium (*Figure 6H–I*, *Figure 6—figure supplement 1G–H*), in contrast to the cPCP proteins. These spatial relationships among Gαi, Par6, and the kinocilium are preserved in the loss- and gain-of *Emx2* function mutants (*Figure 6J–M*, *Figure 6—figure supplement 1I–L*). However, in the GOF-SE late utricles, in which the hair bundle reversal phenotype was only partially penetrant, some of the HCs with normally-positioned kinocilia in the medial utricle showed mislocalization of Gαi and Par6 (*Figure 6N–O"*, *Figure 6—figure supplement 1M–N"*, black arrows with asterisks), whereas Pk2 localization remained unchanged (*Figure 6—figure supplement 2*). Therefore, these results suggest that by delaying ectopic *Emx2* expression until after E15.5, while Emx2 can no longer change the position of kinocilia that has already been established, it can still effectively alter the distribution of Gαi and Par6. This suggests that downstream effectors of Emx2 may normally mediate hair bundle polarity by altering the intracellular polarity complex.

## Blocking Gαi partially rescues the polarity reversal phenotype induced by ectopic *Emx2*

Blocking the Gαi activity with pertussis toxin (Ptx) or knocking out one of the genes that encodes Gαi, *Gnai3,* can lead to misoriented or reversed hair bundle polarity (*Ezan et al., 2013*; *Tarchini et al., 2013*) in cochlear HCs. The polarity-reversal phenotype is rarely observed among cPCP mutants (*Montcouquiol et al., 2003*; *Wang et al., 2005*, *2006*) but resembles those in the *Emx2* mutants. The relocated hair bundle to the same side of the HC in Ptx mutants where the cPCP protein Fz is located, is similar to other epithelial cells in the brain and wing, which we considered to

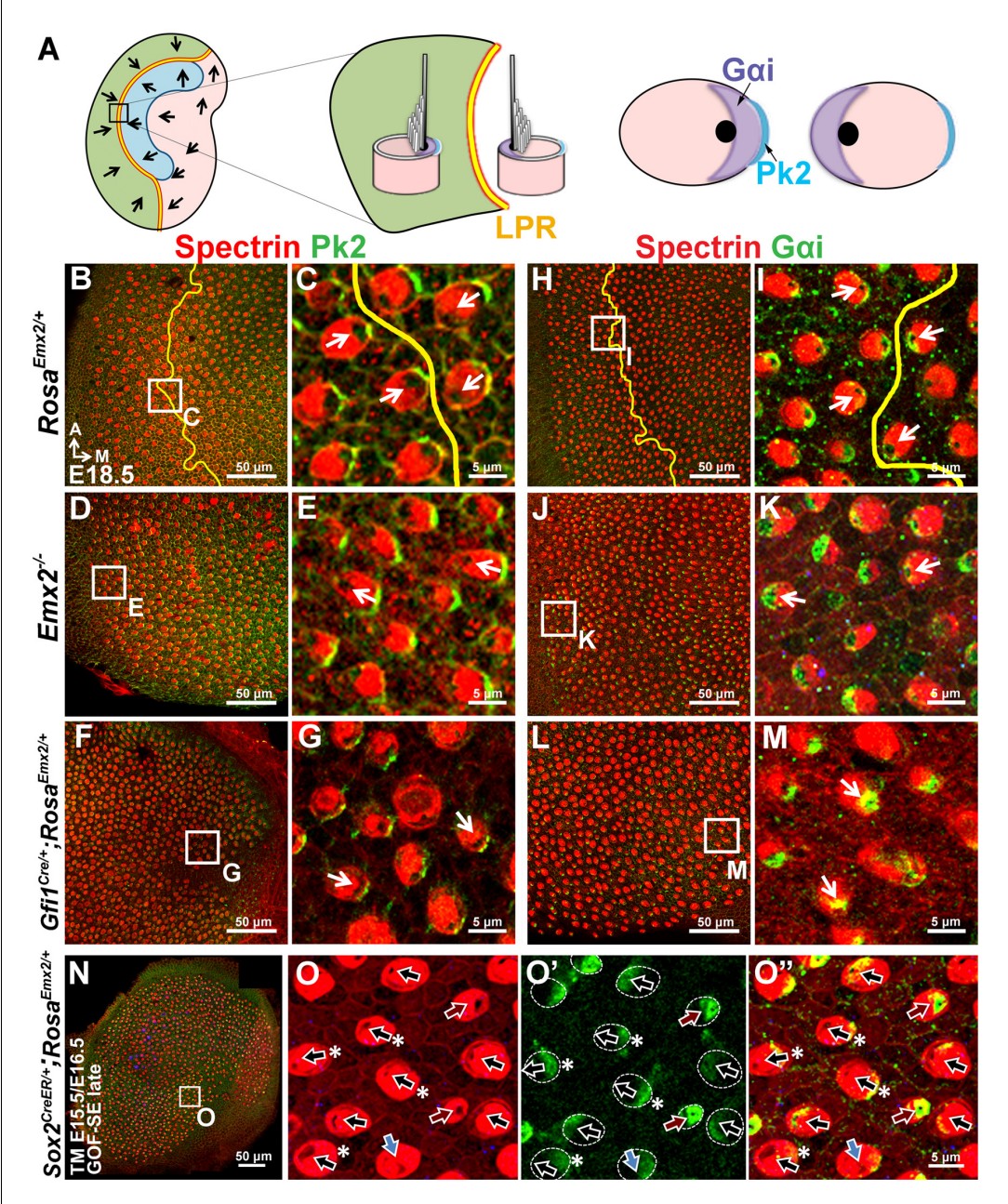

**Figure 6.** Hair bundle polarity reversal is associated with reversed Gαi but not Pk2 localization in *Emx2* mutants. (A) Schematic diagram illustrates the distribution of Pk2 and Gαi in HCs across the LPR of a control utricle. (B–C) In control utricles, Pk2 staining (green) is concentrated on the medial side of HC-supporting cell border across the LPR (n = 15). (D–G) In the lateral region of *Emx2*⁻/⁻ (D-E; n = 5) and medial region of *Gfi1*^Cre/+;*Rosa*^Emx2/+ utricles (F-G; n = 4), in which the hair bundle polarity is reversed compared to the respective left and right side of the LPR in controls (C), Pk2 staining remained on the medial side (green). (H–M) In control utricles (H–I), Gαi immunostaining (green) is associated with the kinocilium. This relationship is consistent among all HCs of the utricle and the staining pattern of Gαi is reversed across the LPR (yellow line) (I; n = 8). A similar spatial relationship between Gαi and the kinocilium position is observed in *Emx2*⁻/⁻ (J-K; n = 4) and *Gfi1*^Cre/+;*Rosa*^Emx2/+ (L-M; n = 5) utricles as controls. (N–O") *Sox2*^CreER/+;*Rosa*^Emx2/+ utricles ( GOF-SE late) show mixed phenotypes including normal (black arrows), misoriented (blue arrows), and reversed (red arrows) kinocilia (n = 3). Some HCs with normal polarity show the abnormal Gαi distribution (asterisks).

The following figure supplements are available for figure 6:

**Figure supplement 1.** Distribution of Vangl2 and Par6 immunoreactivites in apical HCs is not changed in *Emx2* mutant utricles.

**Figure supplement 2.** Distribution of Pk2 immunoreactivites in apical HCs is maintained in *Emx2* GOF-SE late mutant utricles.

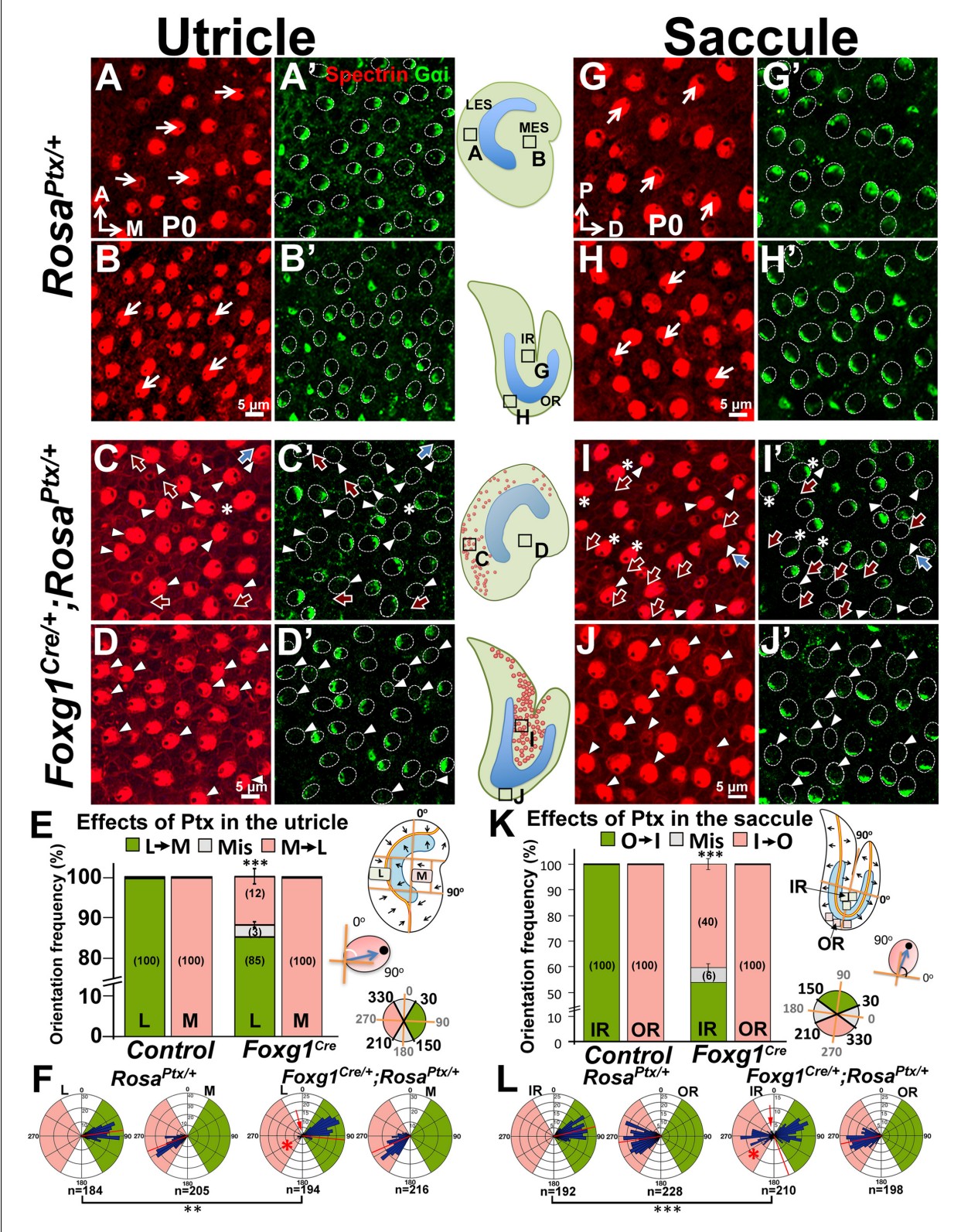

**Figure 7.** Inhibition of Gαi reverses hair bundle orientation in the *Emx2*-positive domain of maculae. (**A–B'**) In control utricles, hair bundles are pointing toward medial and lateral in the respective LES and MES (arrows), and Gαi staining (**A',B'**) is associated with the kinocilium (**A,B**; n = 3). (**C–D'**) In *Foxg1*^*Cre/+*^;*Rosa*^*Ptx/+*^ utricles, hair bundles in the LES (**C**) are sometimes reversed (red arrows) or misoriented (blue arrows) but not in the MES (**D**; n = 3). The accumulation of Gαi in HCs is reduced across the utricle (**C',D'**, arrowheads). Occasionally, uncoupling between the kinocilium and Gαi is found in

*Figure 7 continued on next page*

*Figure 7 continued*

HCs of the LES (C-C', asterisks). (E,F) Quantification of the hair bundle orientation in regions L and M of control and *Foxg1^{Cre/+}; Rosa^{Ptx/+}* utricles are plotted in a bar graph (E; *Figure 7—source data 1*) and circular histogram (F). HCs with the kinocilium positioned within 30°-150° are green (pointing medial), between 210°-330° are pink (pointing lateral), and intermediates (misoriented) are denoted by grey. In region L of mutants, 15% HCs showed abnormal hair bundle orientations. (G–J') Compared to controls (G-H'; n = 3), reduced Gαi is observed across *Foxg1^{Cre/+};Rosa^{Ptx/+}* saccules (I-J'; n = 3) but misoriented (blue arrows) or reversed (red arrows) kinocilia are only found in the IR (I–I') but not the OR (J–J'). (K–L) Quantification of hair bundle orientation pointing toward inner or outer margin are defined as 30°–150° (green) or 210°–330° (pink), respectively (*Figure 7—source data 1*). In the IR of mutants, 46% of hair bundles showed reversed or misoriented polarity. Red asterisks, arrows, and lines represent hair bundle reversal, misorientation, and average degree of HC orientation, respectively. Schematic diagrams indicate the locations of where the corresponding panels were taken as well as positions of all HCs with reversed hair bundles are found in the sample (red dots). **p<0.01, ***p<0.001.

The following source data is available for figure 7:

**Source data 1.** Quantification of hair bundle orientation in *Ptx* mutants.

be the default pattern (*Strutt, 2001*; *Vladar et al., 2012*; *Tree et al., 2002*; *Boutin et al., 2014*; *Guirao et al., 2010*). The bundle polarity phenotype caused by Ptx suggests that heterotrimeric G proteins are normally required to reverse the kinocilium from its default location in the cochlea. Since cochlear HCs express *Emx2* and ectopic *Emx2* is capable of affecting Gαi localization independent of the kinocilium (*Figure 6N–O"*), we hypothesized that Emx2 normally utilizes the Insc/LGN/Gαi complex to change the kinocilium from the default location in cochlear HCs. Under this scenario, Ptx should affect the hair bundle position in macular regions where *Emx2* is expressed and may have no effect in the *Emx2*-negative regions. Additionally, Ptx should block polarity changes induced by ectopic Emx2.

We tested this hypothesis by first examining hair bundle polarity in maculae overexpressing the catalytic S1 subunit of Ptx by crossing *Rosa^{Ptx/+}* mice (*Regard et al., 2007*) with *Foxg1^{Cre/+}* strain, in which *Cre* is activated early at the otic placode stage (*Hébert and McConnell, 2000*). Compared to controls (*Figure 7A–B',G–H'*), hair bundle misorientation (blue arrows) and reversal (red arrows) were observed only in *Emx2*-positive, LES of utricles (*Figure 7C–C',E–F*, 15%) and IR of saccules (*Figure 7I–I',K–L*, 46%) of *Foxg1^{Cre/+};Rosa^{Ptx/+}* ears as predicted by the hypothesis. No apparent phenotype was observed in the *Emx2*-negative medial utricle or the OR of the saccule (*Figure 7D–D',E–F,J–J',K–L*). Despite the preferential hair bundle phenotype in the *Emx2*-positive regions, diffuse or reduced Gαi immunostaining was broadly evident, indicating that Ptx affected the entire mutant maculae (*Figure 7C',D',I',J'*, arrowheads). These results indicate that Ptx affects the distribution of Gαi in macular HCs, similar to cochlear HCs (*Ezan et al., 2013*; *Tarchini et al., 2013*). Additionally, within the *Emx2*-positive domain, some Gαi staining was no longer associated with the kinocilium, regardless of whether the kinocilium location was normal or reversed (*Figure 7C–C',I–I'*, asterisks). The uncoupling of Gαi and kinocilium observed in the *Emx2*-positive domains suggests a stronger effect of Ptx in these regions. More importantly, only the kinocilium positions within the *Emx2*-positive domain were affected by Ptx is consistent with the hypothesis that Ptx only affects the kinocilium position in the regions where *Emx2* is expressed.

Next, we investigated whether hair bundle polarity reversal caused by ectopic *Emx2* can be inhibited by Ptx. Since *Foxg1^{Cre/+};Rosa^{Emx2/Ptx}* compound mutants are early lethal, we generated compound mutants of *Emx2* and *Ptx* using *Gfi1^{Cre/+}* mice. First, HC-specific induction of Ptx in *Gfi1^{Cre/+}; Rosa^{Ptx/+}* utricles (*Figure 8*) showed similar but milder polarity phenotypes than *Foxg1^{Cre/+};Rosa^{Ptx/+}* mutants (*Figure 7*). Only 7% of hair bundle in the LES were affected, whereas hair bundle polarity was normal in the MES of *Gfi1^{Cre/+};Rosa^{Ptx/+}* utricles (*Figure 8C–D',I–K*). Additionally, a similar abnormality in Gαi distribution and its uncoupling from the kinocilium as *Foxg1^{Cre/+};Rosa^{Ptx/+}* maculae was observed (*Figure 8C–D'*, arrowheads and asterisks). Comparing *Ptx* single (*Gfi1^{Cre/+}; Rosa^{Ptx/+}*) with *Ptx* and *Emx2* compound mutant utricles (*Gfi1^{Cre/+};Rosa^{Emx2/Ptx}*) revealed no significant change in hair bundle polarity in the LES (*Figure 8C–C',G–G',I*, 7% versus 9%). However, despite the absence of polarity defect in the medial utricle of *Ptx* mutants (*Figure 8D–D',I–K*), a moderate but significant rescue of bundle polarity reversal induced by ectopic *Emx2* (*Figure 8E–F*) was observed in *Ptx* and *Emx2* compound mutants (*Figure 8H–H'*, yellow arrows, I, 22%). These results suggest that Ptx not only blocks endogenous hair bundle polarity within the *Emx2* domain, it

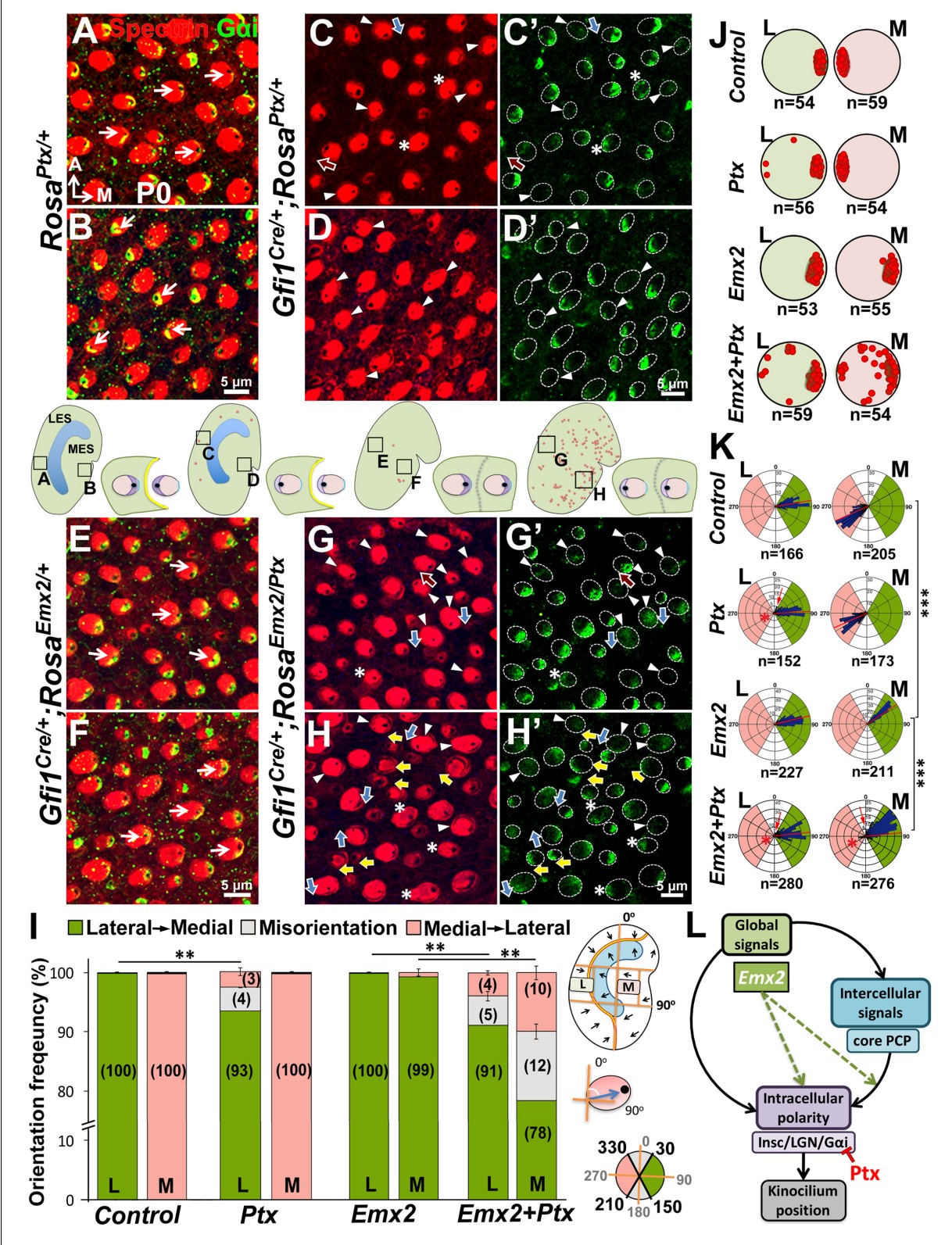

**Figure 8.** *Emx2*-mediated hair bundle polarity reversal requires heterotrimeric G-proteins. (**A–H'**) Compared to *Rosa^Ptx/+* controls (**A–B**), some hair bundles show reversed polarity in the LES (**C–C'**) but not the MES (**D–D'**) of *Gfi1^Cre/+;Rosa^Ptx/+* utricles. Ectopic *Ptx* and *Emx2* in *Gfi1^Cre/+;Rosa^Emx2/Ptx* utricles rescued -bundle polarity reversal in the MES (**H–H'**) but has no additional effect in the LES (**G–G'**). Red, blue and yellow arrows represent reversed, misoriented and rescued hair bundle polarity, respectively. Arrowheads and asterisks represent anti-Gαi staining that is reduced/diffuse or no

*Figure 8 continued on next page*

*Figure 8 continued*

longer associated with the kinocilium, respectively. Schematic diagrams indicate the locations where panels A-H were taken as well as the positions where reversed hair bundles were found (red dots). (I) Quantification of the hair bundle orientation (*Figure 8—source data 1*). Ptx causes abnormal hair bundle polarity in region L of controls (7%) and ectopic *Emx2* (9%). Reversed hair bundle polarity induced by *Emx2* in region M is partially rescued by ectopic *Ptx* (22%, **p<0.01; n = 3 for each group). (J) Plots of kinocilium location (red dots) in HCs within regions L and M of one specimen. (K) Circular histograms summarizing HC orientation distributions in regions L and M for each genotype indicated (n = 3). ***p<0.001. (L) Model of Emx2 effectors in kinocilium positioning by directly regulating the Insc/LGN/Gαi complex (purple) or indirectly regulating the interaction between intercellular (light blue) and intracellular polarity pathways.

The following source data is available for figure 8:

**Source data 1.** Quantification of hair bundle orientation in utricles of *Ptx* single and *Ptx* and *Emx2* compound mutants.

also rescues polarity effects induced by ectopic *Emx2* in regions that is not normally affected by Ptx. Taken together, our results indicate Emx2 requires heterotrimeric G proteins in mediating hair bundle position.

## Conserved role of *Emx2* in establishing hair bundle polarity pattern

Since the LPR is conserved among vertebrates (*Desai et al., 2005*; *Huss et al., 2010*; *Hammond and Whitfield, 2006*), we asked whether *Emx2* has a role in establishing the LPR in other species. We first investigated the chicken inner ear, which has an additional macular organ, the lagena, which exhibits the LPR. Our immunostaining results indicate that Emx2 is expressed in the LES and IR of the respective utricle and saccule in chicken (*Figure 9A–B',D*). In the lagena, its expression is restricted to the region closer to the auditory sensory epithelium, the basilar papilla (*Figure 9C–D*). Consistent with the mouse, the border of the Emx2 expression domain (green line) coincides with the LPR (yellow line) in all three chicken maculae.

The lateral line system is responsible for detecting water pressure changes, which allows aquatic vertebrates to participate in schooling behavior, avoid predators and catch preys (*Chitnis et al., 2012*). This system is made up of clusters of HCs called neuromast. Within each neuromast organ, HCs are arranged in pairs with their hair bundles pointing toward each other, aligned in either anterior-posterior (A-P) or dorsal-ventral (D-V) direction (depending on the neuromast) along the body axis (*Figure 10A–B*; *López-Schier et al., 2004*). Core PCP proteins also regulate hair bundle polarity in neuromasts but their normal distribution is similar between HCs with opposite bundle polarities (*Mirkovic et al., 2012*). These similarities to the maculae prompted us to investigate the role of emx2 in the neuromast.

We found that in zebrafish, emx2 is expressed in half of the HCs, oriented towards the posterior or ventral direction within the respective A-P and D-V neuromast (*Figure 10C–C'''*). To test the role of emx2 in establishing hair bundle polarity in zebrafish, we generated loss- and gain- of *emx2* zebrafish mutants. Using CRISPR/Cas9, we generated *emx2* knockouts (*Figure 10—figure supplement 1A*). In *emx2* knockouts, all hair bundles were uniformly polarized toward the anterior or dorsal direction in the respective A-P and D-V neuromasts (*Figure 10D–D'''*). In contrast, neuromasts of *m6b:emx2-mCherry* transgenic fish (*emx2* gof), which overexpress *emx2* under a HC-specific promoter *myosin6b*, showed hair bundles pointing only toward the posterior or ventral direction in A-P and D-V neuromasts, respectively (*Figure 10E–E'''*). Additionally, in zebrafish utricle and cristae, we found the predicted polarity reversal phenotype in loss- and gain- of *emx2* function larvae comparable to what we observed in mice (*Figure 10—figure supplement 1B–J*). Taken together, these results indicate that Emx2 has an evolutionarily conserved role in determining hair bundle polarity of sensory HCs and establishing the LPR.

Next, we investigated whether the unidirectional hair bundle polarity shown in *emx2* zebrafish mutants exhibit the predicted functional change in directional sensitivity. Specifically, we tested whether all HCs in A-P neuromasts only respond to stimulus from the anterior direction in *emx2* gof fish. We used a fluid jet to stimulate control and *emx2* gof neuromasts expressing *m6b:GCaMP6s-CAAX* in either the anterior or posterior direction and measured mechanically evoked calcium responses (*Figure 10F–J*). Our results showed that in controls, a similar proportion of HCs exhibited mechanically evoked calcium responses from either direction (*Figure 10F–G',J*). In contrast, in *emx2*

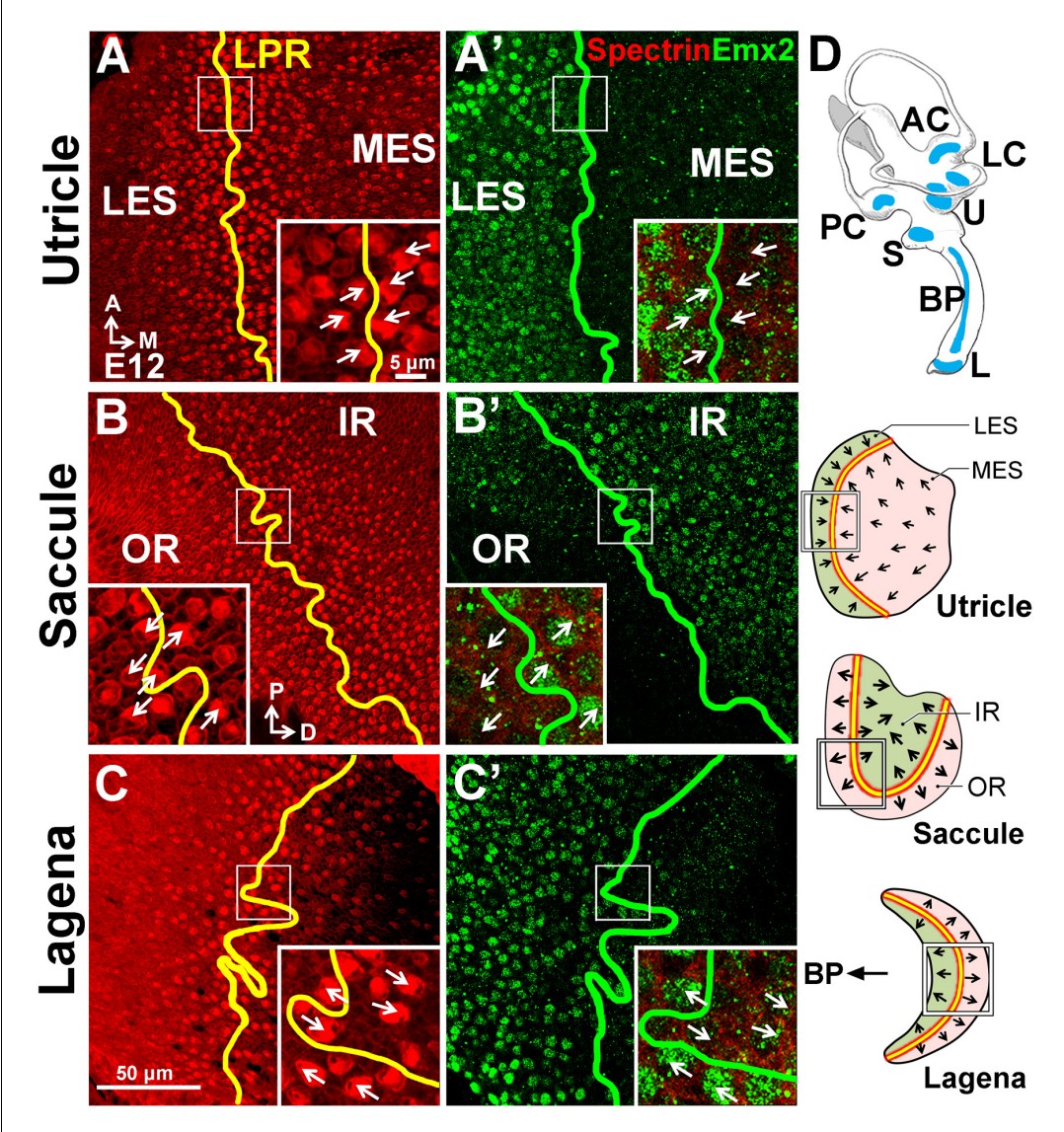

**Figure 9.** Conserved expression of Emx2 in the three chicken maculae. (A–C') Anti-Emx2 immunostaining of the chicken utricle (A,A'), saccule (B,B') and lagena (C,C') at E12. (A–C) Anti-spectrin staining (red) shows opposite hair bundle orientation across the LPR (yellow line). In the same regions shown in (A), (B), and (C) but at the level of the cell body, the border (green line) of the Emx2-positive region (green) is restricted to only one side of the LPR: lateral region of the utricle (A'), inner region of the saccule (B'), and proximal region of the lagena, which is closer to the cochlea/basilar papilla (BP) (C'), respectively (n = 3). (D) Schematic diagrams of the chicken inner ear and its three macular organs.

gof A-P neuromasts, all HCs showed a robust increase in calcium during a stimulus directed towards the posterior, and a corresponding decrease in calcium when the stimulus was directed towards the anterior. Our calcium imaging results show that *emx2* gof HCs only respond to anterior-to-posterior stimuli, which are consistent with the unidirectional posterior-pointing hair bundles in these mutants (*Figure 10H–J*). In support of these - hair bundle polarity defects, the *emx2* gof larvae showed abnormal swimming behavior before their demise. We predict that *Emx2* knockouts in both zebrafish and mouse are likely to change the directional sensitivity of their HCs and exhibit behavioral deficits if viable.

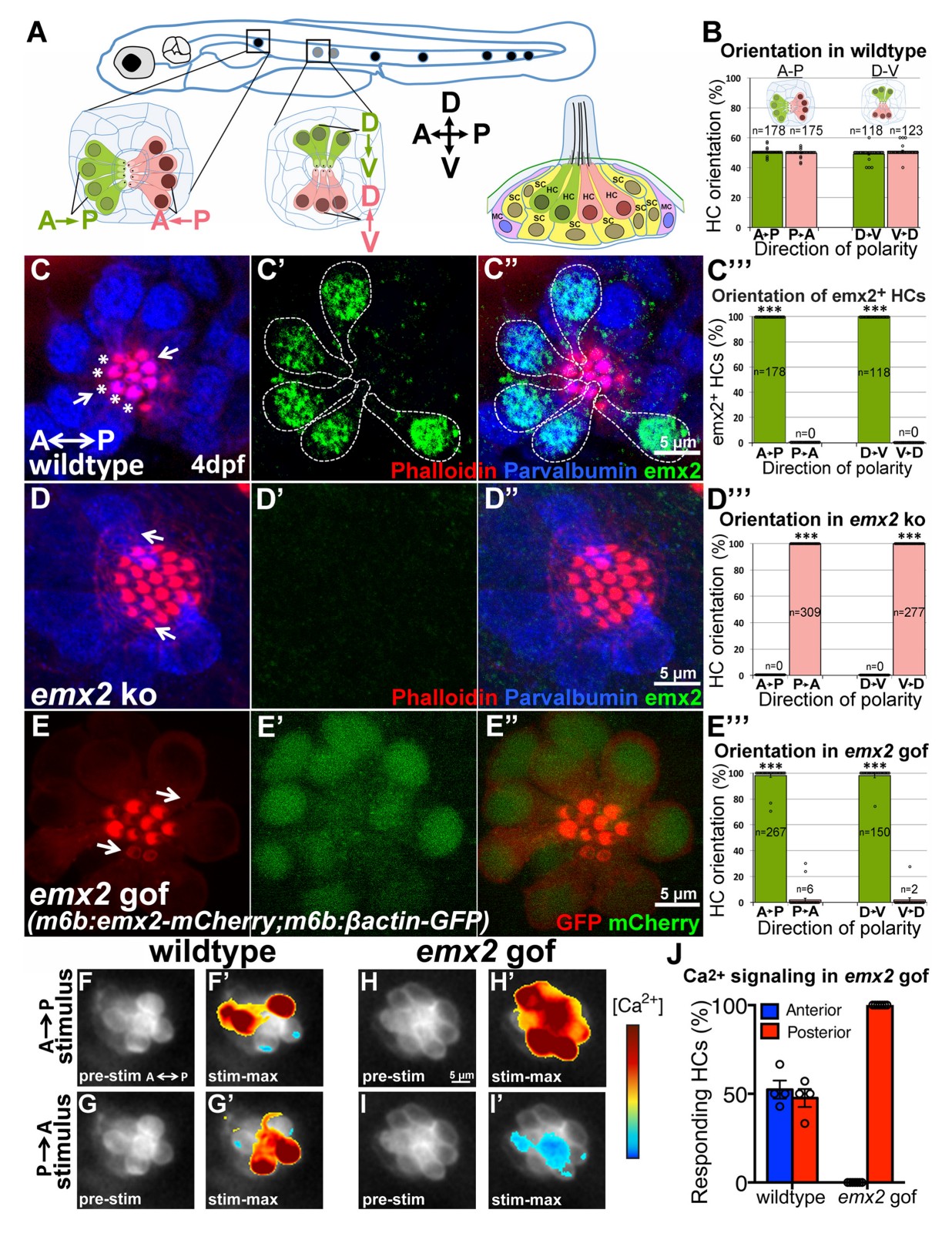

**Figure 10.** *emx2* regulates hair bundle polarity in neuromasts. (**A**) Surface view of an A-P and D-V oriented neuromast and a sagittal view of the cellular architecture of a neuromast in the lateral line of zebrafish. (**B**) HCs with opposite orientation are in a 1:1 ratio. (**C–C"**) An example and (**C'''**) quantification showing only HCs (parvalbumin-positive, blue) with hair bundles (phalloidin, red) oriented toward the posterior (**C**, asterisks) or ventral direction are positive for anti-Emx2 staining (green) in A-P and D-V neuromasts, respectively (55 neuromasts from 26 larvae, ***p<0.001; *Figure 10—*

*Figure 10 continued on next page*

*Figure 10 continued*

*source data 1*). (D–D") An example and (D"') quantification of HCs in *emx2* knockouts showing all HCs are *emx2*-negative (D') and pointing towards the same direction (36 neuromasts, 18 larvae, \*\*\*p<0.001; *Figure 10—source data 1*). (E–E"') All hair bundles (visualized using *m6b:βactin-GFP,* red) are Emx2-positive (E',E'', green) and pointing toward the posterior in the A-P oriented neuromasts of gof mutants. (E'") Quantification of hair bundle polarity in 43 A-P and D-V neuromasts from 23 larvae (*Figure 10—source data 1*). \*\*\*p<0.001. (F–J) In response to an anterior-to-posterior stimulus (F') or posterior-to-anterior stimulus (G') a similar percentage of wildtype HCs show mechanically-evoked increase in calcium levels (J; *Figure 10—source data 2*). *Emx2* gof HCs only respond to an anterior-to-posterior stimulus (H') and are inhibited by a posterior-to-anterior stimulus (I'). The heatmap indicates the change in calcium levels during the first half of the 2s-step stimulus (2 larvae and 4 neuromasts in control, 2 larvae and 7 neuromasts in *emx2* gof). Error bar represents SEM.

The following source data and figure supplement are available for figure 10:

**Source data 1.** Quantification of hair bundle orientation in neuromasts of *emx2* loss- and gain-of-function mutants.

**Source data 2.** Quantification of calcium-activated HCs with oriented stimuli in *emx2* gain-of-function neuromasts.

**Figure supplement 1.** Polarity phenotypes in the utricle and lateral crista of *emx2* loss- and gain-of function zebrafish mutants.

## Discussion

### Emx2 is a global polarity cue

The intrinsic polarity of individual cells within a tissue is regulated by global and intercellular polarity cues, which collectively give rise to the tissue's polarity. The molecular mechanisms of these cross-talks are not well understood (*Figure 8L*). The regional expression pattern of Emx2 in the maculae and its role in reversing hair bundle polarity within its expression domain qualifies this transcription factor as a global polarity cue. Since distribution of the cPCP proteins in maculae is not changed between Emx2-positive and negative regions across the LPR suggests that HCs in the Emx2 domain normally do not undergo cellular rotation to acquire the polarity reversal pattern, as in the case of ommatidia formation in *Drosophila* (*Jenny, 2010*). Furthermore, the normal distribution of cPCP proteins in *Emx2* mutants indicates that *Emx2* effectors function either independently or downstream from the cPCP proteins.

In most other epithelial tissues, trichomes or cilia are associated with the side of the cell where Fz is located and opposite to the side where Vangl and Pk are located (*Strutt, 2001*; *Vladar et al., 2012*; *Tree et al., 2002*; *Boutin et al., 2014*; *Guirao et al., 2010*). A similar pattern is observed in the medial utricle. However, the relationship between distribution of cPCP proteins and hair bundle polarity appears to be reversed in the presence of *Emx2*. For example, hair bundles in *Emx2*-positive domains are located opposite to the Fz expression in cochlear HCs and are associated with Pk in lateral utricular HCs (*Figure 6C,E*; *Deans et al., 2007*; *Montcouquiol et al., 2006*; *Wang and Nathans, 2007*). These results suggest that the interpretation of normal polarity cues is changed by Emx2.

Although Emx2 does not normally function by regulating cPCP proteins but Emx2 changing downstream effectors of cPCP proteins remains a distinct possibility (*Figure 8L*). Regulating downstream effectors that resulted in altered tissue polarity has been reported when a cPCP component, pk, was disrupted in *Drosophila* (*Gubb et al., 1999*). The abnormal ratio of isoforms between *pk* and *spiny-legs (pk:sple)* does not affect the distribution of Dachsous or Fat but changes the effective dose of Dachs (atypical myosin), an effector of the Dachsous-Fat polarity pathway (*Olofsson et al., 2014*; *Ayukawa et al., 2014*; *Ambegaonkar and Irvine, 2015*). Consequently, localization of fz and strabismus (Vangl in mammals) as well as the trichomes are reversed in the wing and abdomen of fly mutants. In a similar manner, Emx2 could regulate downstream effectors of the cPCP proteins and alter the intrinsic hair bundle polarity (*Figure 8L*). The cPCP effectors are known to affect the cilia position in other systems (*Park et al., 2006*; *Wong and Adler, 1993*; *Carvajal-Gonzalez et al., 2016a*, *2016b*) and centrioles/basal body positioning could be the level where the Emx2 effectors alter the interpretation of the intercellular polarity cues.

## Emx2 mediates hair bundle positioning via the heterotrimeric G proteins

As a transcription factor, Emx2 regulates diverse cellular pathways such as regional specification, cell proliferation and fates (*Pellegrini et al., 1996*; *Tole et al., 2000*; *Heins et al., 2001*; *Mallamaci et al., 2000*). Therefore, it is possible that Emx2's effects on hair bundle reversal is indirect, resulting from changing the HC fate or other cellular pathways. Nevertheless, whichever the pathway(s) might be, the outcome is a cell-autonomous switch in the location of the hair bundle by 180°, in part, via the heterotrimeric G proteins (*Figure 8L*). This conclusion is based on several lines of evidence. First, ectopic *Emx2* is sufficient to relocate Gαi and Par6 even after the kinocilium is established suggests that Emx2 effectors regulate intracellular polarity signaling. Second, the ability of Ptx to block hair bundle polarity of Emx2-positive HCs in cochlea and maculae, though variable in efficiency among different mutant strains (*Figures 7–8*; *Ezan et al., 2013*; *Tarchini et al., 2013*), suggest that heterotrimeric G proteins are critical for this process. Third, the ability of Ptx to block ectopic Emx2's effects on polarity further supports the requirement of G proteins by Emx2 effectors. Extrapolating from previous results (*Ezan et al., 2013*; *Tarchini et al., 2013*), Emx2 effectors most likely mediate the change in hair bundle position via the Insc/LGN/Gαi complex. Additionally, our results suggest that this Insc/LGN/Gαi complex is more important for *Emx2*-positive than negative regions and other mechanisms are required for targeting the kinocilium in *Emx2*-negative HCs. Taken together, our results illustrate a global polarity cue bypasses the cPCP proteins and functions at the intracellular level.

## Other roles of Emx2

Our results link Emx2 to basal body positioning in sensory HCs. Notably, Emx2 has been implicated in regulating symmetric versus asymmetric cell division in the brain (*Galli et al., 2002*; *Heins et al., 2001*), which could also be a result of altering spindle orientation. Though speculative, it is possible that Emx2 effectors could function at multiple steps of a cell cycle: spindle orientation during mitosis and/or basal body positioning during differentiation. Identifying these effectors will be important. Furthermore, in both maculae and neuromasts, HCs with opposite hair bundle polarities are innervated by different populations of neurons that project, at least in the maculae, to different regions of the brain (*Maklad et al., 2010*; *Nagiel et al., 2008*; *Pujol-Martí et al., 2014*). These innervation patterns could be guided by downstream targets of Emx2, and Emx2 has also been implicated in mediating neuronal migration in the brain (*Shinozaki et al., 2002*). In summary, our results demonstrated a conserved role of Emx2 in mediating hair bundle polarity in sensory HCs. This largely unexplored cellular process of planar targeting of the cilium at the apical cell surface has profound effects on HC function and may have broader implications in other ciliated cells and neuronal pathfinding.

## Materials and methods

### Mouse strains

The *Rosa*[Emx2-GFP](designated *Rosa*[Emx2]) mouse strain was generated by knocking in the cassette, *attb-pCA* promoter-*lox-stop-lox-Emx2-T2A-Gfp-WPRE-polyA-attb,* to the Rosa locus using integrase technology (conducted by Applied StemCell, Inc., Milpitas, CA, *Tasic et al., 2011*). One founder with the correct insertion, identified based on PCR analyses, was propagated and maintained in the FVB background. Primers used for genotyping offspring are as follow: PR425 (GGTGATAGG TGGCAAGTGGTATTC) and pCA-R2 (GGCTAT GAACTAATGACCCCGT) for the *Rosa*[Emx2-GFP] allele, and R10 (CTCTGCTGCCTCCTGGCTTCT) and R11 (CGAGGCGGATACAAGCAATA) for the wildtype allele with expected fragment sizes of 369 and 311 base pairs, respectively.

*Emx2*[+/-] mice were provided by Peter Gruss at the Max-Planck Institute (RRID:IMSR_EM:00065; *Pellegrini et al., 1996*) and maintained in a mixed C57BL/6J and CD1 background. *Emx2*[Cre] mice were obtained from Shinichi Aizawa at RIKEN Center for Developmental Biology and maintained in the C57BL/6J background (RRID:IMSR_RBRC02272; *Kimura et al., 2005*). *Gfi1*[Cre]knock-in mice were obtained from Lin Gan at University of Rochester (RRID:MGI:4430834; *Yang et al., 2010*]), *Sox2-*[CreER]mice from Konrad Hochedlinger at Harvard University (RRID:IMSR_JAX:017593; *Arnold et al., 2011*), and the *Foxg1*[Cre] mice from Susan McConnell at Stanford University (RRID:IMSR_JAX:

004337; *Hébert and McConnell, 2000*). *Rosa* $^{Ptx/+}$ mice were generated by Shaun Coughlin at University of California at San Francisco (RRID:MGI:3784870; *Regard et al., 2007*) and provided by Yingzi Yang at Harvard Medical School. *Gfi1*$^{Cre}$ and *Sox2*$^{CreER}$ strains were maintained in a CD-1 background, while *Foxg1*$^{Cre}$ strain was maintained in a mixed background of C57BL/6J and Swiss Webster. *Rosa26R*$^{tdTomato}$ (designated *Rosa*$^{tdT}$, RRID:IMSR_JAX:007914, *Madisen et al., 2010*) and *Rosa26R*$^{mtdTomato/mGFP}$ (designated *Rosa*$^{mT/mG}$, RRID:IMSR_JAX:007576, *Muzumdar et al., 2007*) were purchased from Jackson laboratory and maintained in a C57BL/6J background. All animal experiments were conducted under approved NIH animal protocols (#1212-14, #1362-13) and according to NIH animal user guidelines.

## Zebrafish strains

Zebrafish were maintained under standard conditions. All transgenic fish were maintained in a TAB5 wildtype background (S Burgess, NIH). The transgenic line, *Tgmyosin6b:emx2-p2A-nls-mCherry*, designated as *m6b:emx2-mCherry*, was generated using a Gateway cloning Technology. First, a middle entry clone, *pME-emx2*, was constructed using a zebrafish *emx2* cDNA clone (IMAGE: 7403786) with PCR primers encoding attB sites: attB1 *emx2* forward primer (GGGGACAAGTTTGTACAAAAAAG-CAGGCTGCCACCATGTTTCAACCCACACCGAAGAGGTG) and attB2 *emx2* reverse primer (GGGGCACTTTGTACAAGAAAGCTGGGTGATCGTCTGAGGTGACGTCAATTTCCTC). Then, the *myosin6b:emx2-p2A-nls-mCherry* plasmid was constructed using a 5' entry clone containing the HC-specific promoter of *6myosin6b* (*Obholzer et al., 2008*), the middle entry clone *pME-emx2*, a 3' entry clone encoding *p2A*-nuclear localized *mCherry* (*p3E-p2A-nls-mCherry*, gift from Kristen Kwan at the University of Utah), and a destination vector containing the transgensis marker *pcmcl2-gfp* (tol2kit #395; *Kwan et al., 2007*). Larvae injected with this plasmid at the one-cell stage were selected based on the GFP signal within the heart and raised to adulthood. These founders were bred to wildtype or to the previously described transgenic line, *Tgmyo6b:βactin(actb1)-EGFP*$^{vo8}$, designated as *m6b:βactin-GFP* fish (*Kindt et al., 2012*) for hair bundle polarity analyses. Double transgenic fish were identified based on green hair cells (*βactin-GFP)* and mCherry-positive nuclei (*emx2*).

Knockout *emx2* zebrafish were generated using CRISPR/Cas9 technology as described (*Varshney et al., 2015*). Two target sites on Exon1 of *emx2* were chosen for generating guide RNA: GGTAAAACACCTCTTCGGTGTGG and GGCTTTCACTCCAGCGGCAGGGG (*Figure 10—figure supplement 1A*). Genotyping of injected larvae and F1 larvae was conducted by PCR using the primers emx2-fPCR-Fwd (TCACTTAAACTGGGGAATCTTGA) and emx2-fPCR-Rev (GGAGGAGGTAC TGAATGGACTG), followed by subcloning and sequencing of the PCR fragments. F1 generated fish were analyzed for hair bundle polarity between 3 to 5 days post fertilization (dpf).

To create a transgenic line for calcium imaging to examine the directional sensitivity of HCs, a middle entry GCaMP6s-CAAX clone was created. This version of GCaMP6s was modified for zebrafish and has been used previously (*Tabor et al., 2014*). From the tol2 kit, vectors p3E-*polyA* (301) and pDestTol2CG2 (395), were recombined with p5E-*6myosin6b* (*Kindt et al., 2012*), and the middle entry GCaMP6s-CAAX to create *Tgmyosin6b:GCaMP6s-CAAX*, designated as *m6b:GCaMP6s-CAAX*. Then, double transgenic fish expressing *emx2* and GCaMP6s-CAAX was generated by crossing *emx2 gof* with *Tgmyosin6b:GCaMP6s-CAAX* and screened for larvae expressing green HCs (*GCaMP6s-CAAX*) with mCherry-positive nuclei (*emx2*).

## In situ hybridization

Section and whole mount in situ hybridization of the utricle and saccule were performed as described previously (*Morsli et al., 1998*). In situ probes for *Emx2* (*Simeone et al., 1992*), *β-tectorin* (*Rau et al., 1999*), and *Lfng* (*Morsli et al., 1998*) were prepared as described.

## Whole mount immunostaining

In general, E18.5 mouse embryos were harvested and fixed with 4% paraformaldehyde in PBS at 4°C overnight. For anti-Pk2 and anti-Vangl2 staining, hemi-sectioned embryo heads were fixed for 30 min at 4°C. After fixation, samples were washed with PBS and various inner ear sensory organs were dissected, blocked with PBS containing 0.2% Triton-X and 4% donkey serum for 30 min before incubating with primary antibodies diluted in blocking solution overnight at 4°C. Then, samples were

washed extensively with PBS before incubating with secondary antibodies at 1:250 dilutions in blocking solution for 2 hr at room temperature. Samples were mounted with ProLong Gold Antifade (Invitrogen) after extensive washing with PBS and imaged with a Zeiss LSM780 confocal microscope. All low power immunostaining pictures are composite images taken at 40x magnification.

For staining zebrafish larvae, 3–5 dpf embryos were fixed with 4% paraformaldehyde in PBS for 3.5 hr at 4°C. Post-fixed larvae were rinsed with PBS and treated with pre-chilled acetone for 3–5 min at −20°C. Then, larvae were incubated with a blocking solution (2% goat serum, 1% BSA in PBS solution for 2 hr at room temperature), followed by incubation with primary antibodies diluted in PBS with 1% bovine serum albumin (BSA) at 4°C overnight. The next day, larvae were washed four times with PBS for 5 min each before incubating with secondary antibodies at 1:250 dilutions in blocking solution for 2 hr at room temperature. Then, larvae were washed and mounted with Antifade and imaged with a Zeiss LSM780 confocal microscope.

Primary antibodies used in this study are listed as follow: mouse anti-$\beta$II-spectrin (1:500; BD Biosciences Cat# 612562 RRID:AB_399853), rabbit anti-Emx2 (1:250; KO609, Trans Genic, Fukuoka, Japan), rabbit anti-G$\alpha$i (1: 1000; provided by B. Nurnberg; (*Gohla et al., 2007*; *Ezan et al., 2013*), goat anti-oncomodulin (1:250; Santa Cruz Biotechnology Cat# sc-7446 RRID:AB_2267583), rabbit anti-Pard6 (1:250; Santa Cruz Biotechnology Cat# sc-67393 RRID:AB_2267889), mouse anti-parvalbumin (1:5000; Millipore Cat# MAB1572 RRID:AB_2174013), rabbit anti-Pk2 (1:250; *Deans et al., 2007*) and goat anti-Vangl2 (1:250; Santa Cruz Biotechnology Cat# sc-416561 RRID:AB_2213082). In addition, two rabbit polyclonal antibodies, anti-Pk2 and anti-Emx2 were generated for this study (Thermo Fisher Scientific, Waltham, MA) using the synthetic peptide of Pk2 as described (*Deans et al., 2007*) and the full-length mouse Emx2 protein, respectively. Both antibodies were affinity-purified and used at a 1:1000 dilution and show immunostaining patterns indistinguishable from those of the anti-Pk2 and anti-Emx2 described above. Fluorescence-labeled phalloidin (1:50; #F432, Thermo Fisher Scientific, Waltham, MA) was used to visualize actin in the stereocilia on top of sensory HCs.

Secondary antibodies that were used in these studies are listed as follow: Alexa Fluor 405 Donkey anti rabbit IgG (ab175651, Abcam, Cambridge, MA), Alexa Fluor 405 Donkey anti mouse IgG (ab175658, Abcam, Cambridge, MA), Alexa Fluor 405 Donkey anti goat IgG (ab175664, Abcam, Cambridge, MA), Alexa Fluor 488/568/647 Donkey Anti-Rabbit IgG (Thermo Fisher Scientific Cat# A21206 RRID:AB_2535792/Cat# A10042 RRID:AB_2534017/Cat# A-31573 RRID:AB_2536183), Alexa Fluor 488/568/647 Donkey Anti-Mouse IgG (Thermo Fisher Scientific Cat# A-21202 RRID:AB_2535788/Cat# A10037 RRID:AB_2534013/Cat# A-31571 RRID:AB_162542), Alexa Fluor 488/568/647 Donkey Anti-Goat IgG (Thermo Fisher Scientific Cat# A-11055 also A11055 RRID:AB_2534102/Cat# A-11057 RRID:AB_2534104/Cat# A-21447 RRID:AB_2535864), Alexa Fluor 647 Goat Anti-Mouse IgG (Thermo Fisher Scientific Cat# A-21235 RRID:AB_2535804), and Alexa Fluor 568 Goat Anti-Rabbit IgG (Molecular Probes Cat# A-11011 RRID:AB_143157).

## Tamoxifen administration

A stock solution of 30 mg tamoxifen (T5648, Sigma Aldrich, St. Louis, MO) in 1 ml of corn oil was prepared. To avoid premature abortion of fetuses due to tamoxifen, 0.2 mg $\beta$-estradiol (20 mg/ml of ethanol; E8875, Sigma Aldrich, St. Louis, MO) was added per ml of tamoxifen stock solution. On designated gestation days at noon, pregnant females were gavaged with the tamoxifen containing $\beta$-estradiol stock solution at 1 mg/10 g body weight. The morning of a found plug was considered as embryonic day 0.5.

## EdU administration and cell cycle exit analysis

Pregnant mice were injected intraperitonealy with 5-ethynyl-2'-deoxyuridine (EdU; 1 mg/ml solution; Thermo Fisher Scientific, Waltham, MA) three times (10 am, 12 pm and 2 pm) on a given day between embryonic day (E) 11.5 and E16.5 at an amount of 10 mg EdU/g of body weight, and all embryos were harvested and fixed with 4% paraformaldehyde at E18.5 (*Figure 3G*). EdU-labeled cells were detected with a Click-iT reaction (*Bok et al., 2013*; Thermo Fisher Scientific, Waltham, MA). Processed utricles were then flat-mounted and imaged using LSM780 confocal microscopy.

Under this injection regiment, HC precursors that underwent terminal mitosis soon after EdU incorporation should be strongly labeled, whereas precursors that have already exited from the cell

cycle at the time of EdU delivery should not be labeled. HC precursors that underwent several rounds of cell division after EdU incorporation should have weak or no EdU labeling in the nucleus. Since weak EdU labeling can also be a result of HC precursors that incorporated EdU at the end of S phase of the last cell cycle, we scored all Myosin VIIa-positive HCs that have robust or distinct EdU labeling in the nuclei (*Figure 3I*). For quantification of EdU-labeled HCs, the utricle was divided into three regions: striola, lateral and medial extrastriolar regions (LES and MES). Oncomodulin-positive region was marked as striola. The separation between LES and MES was defined by drawing a straight line linking the two ends of oncomodulin-positive striolar region to the edge of the utricle as shown in *Figure 3H*. Approximately 300, 450, and 500 HCs were counted in the striola, LES and MES of each utricle, respectively.

## Quantification and statistical analyses

For quantification of HC density and surface area in the utricle, a straight A-P line between the two widest points of a given utricle along the anterior-posterior axis was drawn on a stitched confocal image taken at 40x magnification. Two lines perpendicular to the A-P line, marking the middle-third region were drawn and this region was sub-divided into four equal parts marked as 1, 2, 3, and 4 (*Figure 4O*). Within regions 1 and 3, total number of HCs per 0.01 mm$^2$ and apical surface area of HCs were scored as representatives for the lateral and medial region of the utricle, respectively.

For hair bundle orientation analyses, regions 1 and 3 were further divided into three equal sections. Hair bundle angles were measured in the middle one-third, defined as regions L (lateral) and M (medial, *Figure 4P*). Each region contains at least 50 HCs. The hair bundle angle of each HC was measured based on the position of the kinocilium on the apical surface by defining 0° as the anterior apex of the utricle and the medial side as 90°. Since LPR falls within region L of controls, we identified two different groups of HCs in controls (*Figure 4Q*).

In *Ptx* alone or *Emx2* and *Ptx* epistatic experiments of utricles, we defined HC polarity between 30-150° as pointing medial (*Figures 7E–F* and *8I,K*, green) and between 210-330° as pointing lateral (*Figures 7E–F* and *8I,K*, pink). Polarities of hair bundles that are outside of these ranges are considered misorientated (*Figures 7E* and *8I*, grey, and *Figures 7F*, *8K*, white). In order to avoid confusion in accessing polarity phenotypes of mutants, oncomodulin-positive HCs in control and Ptx specimens of region L, in which HCs are pointing toward the lateral edge were not included in the quantification of hair bundle angles shown in regions L of *Figures 7E–F and* and *8I–K*.

To quantify the hair bundle orientation in the saccule, a line drawn along the notch of the saccule was defined as the anterior-posterior axis. Then a perpendicular dorsal-ventral line was drawn, which bisected the A-P line into two halves. We defined the dorsal end as 0° and the posterior end as 90°. Three 50 µm$^2$ squares in the anterior region of the IR and OR were selected and hair bundle polarity within were scored (*Figure 7K*). These regions were selected to avoid variation in polarity among HCs of controls and to specifically exclude the striola, which is bisected by the LPR in the saccule (*Figures 1B,H* and *2P*). At least 50 HCs were counted from each region. hair bundle polarity in the IR and OR of a normal saccule is between 30°–150° (*Figure 7K–L*, green) and 210°−330° (*Figure 7K–L*, pink), respectively. The hair bundle polarity of HCs outside of these two ranges is considered misorientated (*Figure 7K*, grey and *7L*, white).

Statistical analyses of our quantification were performed using Prism 5 (GraphPad Soft-ware). HC density, apical surface area of HC, and hair bundle orientation were analyzed using an unpaired Student's t test or one-way ANOVA with the appropriate post hoc test.

## Calcium imaging

For calcium imaging, measurements were made as previously described (*Kindt et al., 2012*; *Zhang et al., 2016*). Briefly, larvae were anesthetized with 0.03% 3-amino benzoic acid ethylester (MESAB, Western Chemical, Ferndale, WA) in E3, and mounted with tungsten pins onto a Sylgard recording chamber. Larvae were then microinjected in the heart with 125 µM α-bungarotoxin (Tocris, Bristol, UK) to suppress muscle activity. After paralysis, calcium imaging was performed in extracellular solution in mM: 130 NaCl, 2 KCl, 2 CaCl$_2$, 1 MgCl$_2$ and 10 HEPES, pH 7.3, 290 mOsm. A pressure clamp (HSPC-1, ALA Scientific, New York, NY) attached to a glass pipette (tip diameter ~30–50 µm) was filled with extracellular solution and used to mechanically stimulate HCs along the anterior-posterior axis of the fish. Calcium measurements were made on a Nikon Eclipse NiE microscope using a

60 × 1.0 NA CFI Fluor water-immersion objective and the following filter set: excitation: 480/30 and emission: 535/40. The microscope was equipped with an Orca D2 camera (Hamamatsu, Hamamatsu City, Japan), and images were acquired using Elements software (Nikon Instruments Inc., Melville, NY).

## Acknowledgements

We thank Drs. Shinichi Aizawa, Shaun Coughlin, Lin Gan, Peter Gruss, Konrad Hochedlinger, Susan McConnell and Yinzi Yang for mice, Dr. Shawn Burgess for zebrafish, Dr. Bernd Nurnberg for anti-Gαi antibody, and Timothy Chang and Claire Wong for quantification of kinocilium locations. We are grateful to Drs. Ajay Chitnis (NICHD), Cecilia Moens (Fred Hutchinson Cancer Research Center) and investigators at NIDCD including Lisa Cunnigham, Dennis Drayna, Thomas Friedman as well as members of the Wu lab for critical reading and suggestions of the manuscript.

## Additional information

### Funding

| Funder | Author |
| --- | --- |
| National Institute on Deafness and Other Communication Disorders | Katie Kindt<br>Doris K Wu |

The funders had no role in study design, data collection and interpretation, or the decision to submit the work for publication.

### Author contributions

TJ, Data curation, Formal analysis, Validation, Investigation, Methodology, Writing-original draft; KK, Conceptualization, Resources, Formal analysis, Supervision, Funding acquisition, Investigation, Methodology, Writing—original draft, Project administration, Writing—review and editing; DKW, Conceptualization, Resources, Supervision, Funding acquisition, Writing—original draft, Project administration, Writing—review and editing

### Author ORCIDs

Katie Kindt, http://orcid.org/0000-0002-1065-8215
Doris K Wu, http://orcid.org/0000-0002-1400-3558

### Ethics

Animal experimentation: All animal experiments were conducted under approved NIH animal protocols (#1212-14, #1362-13) and according to NIH animal user guidelines.

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
