## [Decision Letter]

Thank you for submitting your manuscript "Transcription factor Emx2 to *eLife*. Your article has been reviewed by three peer reviewers, and the evaluation has been overseen by a Reviewing Editor and a Senior Editor. The reviewers have opted to remain anonymous. As you will see, all of the reviewers were impressed with the importance and novelty of your work. I was, too.

I am including the three reviews (lightly edited) at the end of this letter, as there are a variety of specific and useful suggestions in them.

Reviewer #1:

In the manuscript "Transcription factor Emx2 controls stereocilia polarity of sensory hair cells" by Tao Jiang and colleagues, the role of the transcription factor Emx2 is analyzed within the context of the polarization of mechanosensory hair cells in the murine inner ear, as well as the ear of the chicken and the ear and lateral line of zebrafish.

This manuscript tackles an important question, and a problem that has remained puzzling. It is well written and generally of very high quality. Overall, I believe that it will have a major impact and be of general interest. I support its publication.

Reviewer #2:

This manuscript describes the role of Emx2 in the development of planar polarity and specifically the role of Emx2 in coordinating the orientation of polarized stereociliary bundles about a line of polarity reversal (LPR) in the utricular and saccular maculae. This effort builds upon a prior study of Emx2 mutants that showed loss of the LPR and vestibular hair cells with bundles oriented in a single, uniform direction (Holley et al). The current work provides an in-depth analysis of Emx2 expression, characterization of a complementary overexpression model, and establishes a regulatory connection with the Gαi/Par signaling pathway. Based upon this body of work Emx2 is proposed to function as a global regulator of planar polarity, and principles of this function are shown to be conserved in zebrafish and chick. This is an important contribution to the planar polarity and inner ear research fields. Unfortunately, the figures are assembled in a very dense style which in some instances is a disservice to the complicated and beautiful anatomy of the system and may be difficult to interpret by readers outside of the inner ear field.

1) inconsistencies between *Emx2^Cre^* and stereociliary bundle orientation

*Emx2^Cre^* lineage tracing data presented in Figure 2 each reveal hair cells with stereociliary bundle orientations that match the lateral region of the utricle (2C) or the inner region of the saccule (2K), but do not show evidence of *Emx2^Cre^* activity. This contradicts an untested theme of the manuscript that Emx2 (b/c it is a transcription factor) functions autonomously within cells to direct stereociliary bundle orientation.

An alternative interpretation raised by this lineage tracing data is that Emx2 functions non-cell autonomously. – Perhaps by regulating the expression of a short range secreted factor that is sufficient to initiate all of the cell polarity functions attributed to Emx2. At a minimum, the difference between *Emx2^Cre^* lineage tracing and stereociliary bundle orientation should be a point of discussion. Preferably experiments can be conducted to test the autonomy of the Emx2 mutant phenotype or whether this difference is corrected postnatally.

2) No evaluation of maculae size or area

In the original Holley et al. paper it was reported that, in addition to the changes in stereociliary bundle orientation, there was a reduction in the overall area of the sensory epithelia of the saccule. This could be interpreted as evidence that this region of the saccule is specified by Emx2, and as a result is missing in Emx2 mutant thereby yielding a smaller saccule. The authors could address this lingering interpretation of the Holley et al. study, that Emx2 has a role in specification (in addition to or rather than global polarity), by measuring sensory epithelia area for the utricle and saccule in these Emx2 mutant lines. This would be a complement to the EDU experiments already presented.

3) Incomplete analysis of late-stage Emx2 overexpression (*Sox2^CreERt2^, Rosa^Emx2/+^*) experiments

Based upon experiments in which Emx2 expression is activated at later stages of development it is proposed that Emx2 regulates stereociliary bundle orientation by directing the polarized distribution of Gαi/Par6 relative to the core PCP axis. In these experiments Emx2 is proposed to rapidly alter Gαi or Par6 distribution but occurs too late to alter stereociliary bundle orientation. However, in each figure there are also hair cells with misoriented stereociliary bundles that do not appear to have either a lateral or a medial orientation (blue arrows). Investigating the basis of these hair cells further may provide mechanistic insight.

One possibility is that the distribution of PCP proteins are also changed in these cells and that these bundles remain properly oriented along this new PCP protein axis – independent of Gαi. This could be evaluated by looking at Pk2 distribution in *Sox2^Cre^, Rosa^Emx2+^* tissue. A second possibility is that the stereociliary bundles are dynamically reorienting in response to the Emx2-dependent changes in Gαi or Par6 distributions, and that hair cells represented by blue arrows in these figures are actually in a transition state. This latter point could be evaluated by looking at a later stage when reorientation might be complete.

Other concerns:

1) For Figure 2, this image should span the *Emx2^Cre^* boundary (similar to 2C, 2G and 2K) in order to demonstrate these hair cells have the same orientation in Emx2 mutants.

2) It is very difficult to see the polarity of hair cells in Figure 7. Are the overview images necessary and if removed could the hair cell images be enlarged?

3) In Figure 3 black boxes obscure the lateral region of the utricle in several panels.

4) Several instances where triple labeling is presented that might not be critical for presentation or data interpretation. Simplifying the figures might make the data more accessible. For example, 1E' and 1H' do not need the red channel, similarly in Figure 3 the relative distribution of EDU and the striola could be easier to see without myosin VII labeling (red, 3A', 3D', etc.)

5) Text and grammatical errors are present throughout.

A) 'Stereocilia bundle' is not the correct term for this structure. It is either a 'bundle of stereocilia' or a 'stereociliary bundle'.

B) 'Stereocilia polarity' is similarly not an accurate term. Individual stereocilia do not show planar polarity and instead are polarized based upon actin filament organization and the position of the barbed end. The organization that is being referred to in the manuscript is the 'polarity of the stereociliary bundle' and it should be described as such.

Reviewer #3:

The manuscript by Jiang et al. reports the identification of the transcription factor Emx2 as a global regulator of hair bundle orientation in vertebrates. Specifically, hair bundles of sensory hair cells in vertebrate maculae and in zebrafish neuromasts adopt mirror-image polarity along the line of polarity reversal (LPR). Polarity reversal in one domain was found to correlate with Emx2 expression in all species examined (mouse, chick and zebrafish). The authors then performed a series of elegant genetic manipulations in mice and zebrafish to demonstrate that Emx2 is both necessary and sufficient to pattern and re-pattern hair bundle polarity in a cell-autonomous manner. Importantly, the effect of Emx2 on hair bundle polarity was not due to a loss of the Emx2 expressing cell lineage or altered timing of cell cycle exit in the affected region, indicating a specific effect on polarity establishment. At a mechanistic level, the authors provided evidence that Emx2 mediates polarity reversal in part through heterotrimeric G protein signaling. Interestingly, G protein signaling is not required for the "default" orientation of hair bundles in regions where Emx2 is not expressed, suggesting the existence of additional polarity mechanisms. Finally, calcium imaging in zebrafish demonstrated that Emx2-mediated polarity reversal is important for neuromast hair cells' directional response to mechanical stimuli.

While inter- and intra-cellular Planar Cell Polarity (PCP) signaling mechanisms have been extensively studied, a long-standing question of PCP regulation is how positional information along the body axis generates global patterns of PCP. Through thorough loss- and gain-of-function analyses of Emx2 in both mice and zebrafish, this study convincingly demonstrated that Emx2 plays a key role in dictating global patterns of hair cell PCP in the ear and lateral line. While Emx2 target genes important for this process remain to be determined, they likely act to control basal body positioning through the hair cell-intrinsic polarity machinery. Thus, these findings provide significant new insights into the establishment of global PCP patterns essential for the proper function of these sensory end organs.

I have a couple of suggestions to help strengthen the conclusions of the manuscript:

1) Figure 4 and related text. In addition to the cell-autonomous effect of Emx2 OE (overexpression), which was nicely shown using the *Gfi1^Cre^* driver, could the authors comment on whether there was any non-autonomous effect on re-orienting the hair bundles upon mosaic Emx2 OE driven by *Sox2^CreER^* (i.e. were there any GFP negative hair cells being repolarized)? This would be informative in gauging whether Emx2 had any role in regulating intercellular PCP signaling. It is difficult to see which hair cells are GFP positive in Figure 4. It would be helpful to enhance the signals of the GFP channel.

2) Figure 4. Did Emx2 OE in hair cells result in loss of oncomodulin+ type I hair cells in the utricle? Figure 5 showed that oncomodulin expression was not affected by Emx2 OE in saccular hair cells. It would be nice to also show that for the utricle, to further demonstrate the specific effect of Emx2 on hair bundle polarity.

---

## [Author Response]

Reviewer #2:

[...]

1) inconsistencies between Emx2^Cre^ and stereociliary bundle orientation

Emx2^Cre^ lineage tracing data presented in Figure 2 each reveal hair cells with stereociliary bundle orientations that match the lateral region of the utricle (2C) or the inner region of the saccule (2K), but do not show evidence of Emx2^Cre^ activity. This contradicts an untested theme of the manuscript that Emx2 (b/c it is a transcription factor) functions autonomously within cells to direct stereociliary bundle orientation.

An alternative interpretation raised by this lineage tracing data is that Emx2 functions non-cell autonomously. Perhaps by regulating the expression of a short range secreted factor that is sufficient to initiate all of the cell polarity functions attributed to Emx2. At a minimum, the difference between Emx2^Cre^ lineage tracing and stereociliary bundle orientation should be a point of discussion. Preferably experiments can be conducted to test the autonomy of the Emx2 mutant phenotype or whether this difference is corrected postnatally.

We thank the reviewer for pointing out those cells in Figure 2 that show reversed polarity but no ere reporter activity, which we have inadvertently neglected to explain. Those cells with reversed bundle polarity in Figure 2, though negative for cre reporter activity, are indeed positive for Emx2 immunostaining. We took advantage of the fact that Figure 1 and Figure 2 are taken from the same specimen, and we are now showing identical regions in Figure 1 as Figure 2 to illustrate this relationship. In the utricle shown in Figure 2, we counted a total of only 18 hair cells that are immediately lateral to the LPR with reversed polarity and no cre-reporter activity, but these cells are invariably positive for Emx2 immunostaining. In addition, there are also a total of 21 hair cells in this specimen that are immediately medial to the LPR, which show the default polarity and are positive for cre-reporter activity but lack Emx2 immunoreactivity in the nucleus. A similar pattern is observed in the saccule except there are many more cre-positive hair cells with normal polarity but negative for Emx2 staining in the posterior-ventral region(numbers provided in the legend of Figure 2). Together, these results indicate that Emx2 immunostaining is a much better indicator of hair bundle polarity than the Emx2-lineage reporter (subsection “The LES and IR regions are preserved in Emx2^-/-^ maculae”, first paragraph and Figure 2—figure supplement 1).

A model of non-cell autonomous effect of Emx2 in reversing bundle polarity would predict at least some hair cells with reverse bundle polarity that are negative for Emx2 staining. This was not the case. Hair bundle polarity reversal correlates extremely well with Emx2 immunoreactivities.To this end, we added data from another 2 samples in Figure 2—figure supplement 1 to illustrate this fact.

2) No evaluation of maculae size or area

In the original Holley et al. paper it was reported that, in addition to the changes in stereociliary bundle orientation, there was a reduction in the overall area of the sensory epithelia of the saccule. This could be interpreted as evidence that this region of the saccule is specified by Emx2, and as a result is missing in Emx2 mutant thereby yielding a smaller saccule. The authors could address this lingering interpretation of the Holley et al. study, that Emx2 has a role in specification (in addition to or rather than global polarity), by measuring sensory epithelia area for the utricle and saccule in these Emx2 mutant lines. This would be a complement to the EDU experiments already presented.

We did not observe a change in the size of the maculae or total number of hair cells in mutant utricles. Those results are now in [Supplementary-material SD1-data] and [Supplementary-material SD2-data].

3) Incomplete analysis of late-stage Emx2 overexpression (Sox2^CreERt2^, Rosa^Emx2/+^) experiments

Based upon experiments in which Emx2 expression is activated at later stages of development it is proposed that Emx2 regulates stereociliary bundle orientation by directing the polarized distribution of Gαi/Par6 relative to the core PCP axis. In these experiments Emx2 is proposed to rapidly alter Gαi or Par6 distribution but occurs too late to alter stereociliary bundle orientation. However, in each figure there are also hair cells with misoriented stereociliary bundles that do not appear to have either a lateral or a medial orientation (blue arrows). Investigating the basis of these hair cells further may provide mechanistic insight.

One possibility is that the distribution of PCP proteins are also changed in these cells and that these bundles remain properly oriented along this new PCP protein axis – independent of Gαi. This could be evaluated by looking at Pk2 distribution in Sox2^Cre^, Rosa^Emx2+^ tissue.

The specimen shown in Figure 6 was not processed for anti-PK2 staining, we have now added another specimen in Figure 6—figure supplement 2 showing anti-Pk2 staining after tamoxifen treatment at E15.5 and E16.5. There is no change in the distribution of Pk2 in hair cells with reversed or misoriented bundles, supporting our interpretation that Emx2 functions independently from the cPCP proteins.

A second possibility is that the stereociliary bundles are dynamically reorienting in response to the Emx2-dependent changes in Gαi or Par6 distributions, and that hair cells represented by blue arrows in these figures are actually in a transition state. This latter point could be evaluated by looking at a later stage when reorientation might be complete.

Unfortunately, we cannot evaluate the polarity of misoriented hair cells later than the harvesting ages of E18.5 because the *Sox2^Cre^, Rosa^Emx2+^* (now relabeled as GOF SE late) mice don't survive at birth. However, we do know that percentages of reversed or misoriented hair bundles in the utricles were even lower when tamoxifen was administered postnatally instead of E15.5 and E16.5, confirming that the ability of hair cells to respond to Emx2 is age-dependent.

Other concerns:

1) For Figure 2, this image should span the Emx2^Cre^ boundary (similar to 2C, 2G and 2K) in order to demonstrate these hair cells have the same orientation in Emx2 mutants.

Corrected.

*2) It is very difficult to see the polarity of hair cells in Figure 7. Are the overview images necessary and if removed could the hair cell images be enlarged?*We have incorporated the reviewer's suggestions in the revised manuscript for both Figure 7 and Figure 8 by replacing the low power views with schematic diagrams, which allowed more room to show higher magnifications of the hair cells.

3) In Figure 3 black boxes obscure the lateral region of the utricle in several panels.

Corrected.

4) Several instances where triple labeling is presented that might not be critical for presentation or data interpretation. Simplifying the figures might make the data more accessible. For example, 1E' and 1H' do not need the red channel, similarly in Figure 3 the relative distribution of EDU and the striola could be easier to see without myosin VII labeling (red, 3A', 3D', etc.)

Yes, the revised Figure 3 is much clearer now with the suggested changes. However, we feel the red channel in Figure 1 helps to better delineate the boundary of individual hair cells and we kept it.

5) Text and grammatical errors are present throughout.

A) 'Stereocilia bundle' is not the correct term for this structure. It is either a 'bundle of stereocilia' or a 'stereociliary bundle'.

For simplification, we have changed the wording to hair bundle in most of the text.

B) 'Stereocilia polarity' is similarly not an accurate term. Individual stereocilia do not show planar polarity and instead are polarized based upon actin filament organization and the position of the barbed end. The organization that is being referred to in the manuscript is the 'polarity of the stereociliary bundle' and it should be described as such.

We now use the term hair bundle polarity.

Reviewer #3: […] I have a couple of suggestions to help strengthen the conclusions of the manuscript:

1) Figure 4 and related text. In addition to the cell-autonomous effect of Emx2 OE (overexpression), which was nicely shown using the Gfi1^Cre^ driver, could the authors comment on whether there was any non-autonomous effect on re-orienting the hair bundles upon mosaic Emx2 OE driven by Sox2-CreER (i.e. were there any GFP negative hair cells being repolarized)? This would be informative in gauging whether Emx2 had any role in regulating intercellular PCP signaling. It is difficult to see which hair cells are GFP positive in Figure 4. It would be helpful to enhance the signals of the GFP channel.

In Emx2 OE driven by Sox2-creER (now labeled as GOF-SE) specimens, GFP is expressed in the entire sensory domain, including both hair cells and supporting cells and we cannot definitively identify any GFP negative hair cells. See Figure 11 showing only the green channel for Figure 4D-F and G-I. These data are not informative and we decided not to include them in the manuscript. However, please refer to our response to reviewer#2 item#1 for the lack of evidence for non-cell autonomous effect of Emx2 in altering hair bundle polarity.

Author response image 1.**DOI:**
http://dx.doi.org/10.7554/eLife.23661.028

*2) Figure 4. Did Emx2 OE in hair cells result in loss of oncomodulin+ type I hair cells in the utricle? Figure 5 showed that oncomodulin expression was not affected by Emx2 OE in saccular hair cells. It would be nice to also show that for the utricle, to further demonstrate the specific effect of Emx2 on hair bundle polarity*.

The presence of oncomodulin in the Emx2 OE samples is variable but is consistent between the two maculae. For example, in Figure 4, the oncomodulin staining in the utricle (Figure 4) is sparsely labeled. A similar pattern is observed in the saccule of that specimen (not shown). In contrast, oncomodulin staining is clear in the saccule shown in Figure 5 as well as the utricle (not shown). However, this is the only specimen that shows good antioncomodulin staining out of the five that were processed in a similar manner. These results indicate that ectopic Emx2 does have other effects on hair cells that do not normally express this gene and this effect may be genetic background dependent. This information has been added in the text (subsection “Ectopic Emx2 reverses hair bundle polarity”, first paragraph).